# Toxin-mediated depletion of NAD and NADP drives persister formation in a human pathogen

Isabella Santi[1,4], Raphael Dias Teixeira [ID][1,4], Pablo Manfredi[1], Hector Hernandez Gonzalez [ID][1], Daniel C Spiess [ID][1], Guillaume Mas [ID][1], Alexander Klotz [ID][1,3], Andreas Kaczmarczyk [ID][1], Simon van Vliet[1], Nicola Zamboni [ID][2], Sebastian Hiller [ID][1] & Urs Jenal [ID][1✉]

## Abstract

**Toxin–antitoxin (TA) systems are widespread in bacteria and implicated in genome stability, virulence, phage defense, and persistence. TA systems have diverse activities and cellular targets, but their physiological roles and regulatory mechanisms are often unclear. Here, we show that the NatR–NatT TA system, which is part of the core genome of the human pathogen *Pseudomonas aeruginosa*, generates drug-tolerant persisters by specifically depleting nicotinamide dinucleotides. While actively growing *P. aeruginosa* cells compensate for NatT-mediated NAD+ deficiency by inducing the NAD+ salvage pathway, NAD depletion generates drug-tolerant persisters under nutrient-limited conditions. Our structural and biochemical analyses propose a model for NatT toxin activation and autoregulation and indicate that NatT activity is subject to powerful metabolic feedback control by the NAD+ precursor nicotinamide. Based on the identification of *natT* gain-of-function alleles in patient isolates and on the observation that NatT increases *P. aeruginosa* virulence, we postulate that NatT modulates pathogen fitness during infections. These findings pave the way for detailed investigations into how a toxin–antitoxin system can promote pathogen persistence by disrupting essential metabolic pathways.**

**Keywords** Persisters; *Pseudomonas aeruginosa*; Toxin–antitoxin System; NADase; RES Domain
**Subject Categories** Microbiology, Virology & Host Pathogen Interaction; Structural Biology

## Introduction

Antibiotic tolerance is being recognized as an important survival strategy that bacterial pathogens use to escape the lethal effects of bactericidal antibiotics (Fridman et al, 2014; Brauner et al, 2016; Santi et al, 2021). Subpopulations of drug-tolerant bacteria, called persisters, promote pathogen survival and recurrent infections and provide protected reservoirs for the development of antibiotic resistance (Santi et al, 2021; Levin-Reisman et al, 2019, 2017). Although the molecular basis of persisters is still poorly understood, it is assumed that they can originate from small subpopulations of metabolically dormant cells in response to nutrient exhaustion or to other forms of stress (Kaplan et al, 2021; Brauner et al, 2016). Generally, persisters account for only a small fraction of bacterial populations, making mechanistic studies of drug tolerance challenging. However, hyper-persister lineages evolve during chronic infections (Santi et al, 2021; Mulcahy et al, 2010; Ramsey et al, 1993) or in laboratory populations repeatedly exposed to antibiotics (Van den Bergh et al, 2016; Santi et al, 2021; Fridman et al, 2014), offering entry points into this phenomenon by providing access to larger persister populations and by exposing potential drivers of drug tolerance.

We have recently isolated hyper-persister variants of *P. aeruginosa*, an important human pathogen causing acute and chronic infections (Santi et al, 2021). One of the mutations conferring drug tolerance was mapped to *PA1030*, a gene encoding a RES domain containing toxin that is part of a type II toxin–antitoxin (TA) module (Jurėnas et al, 2022). Its upstream neighbor, *PA1029*, encodes a small putative transcription factor with an N-terminal HTH domain and a C-terminal Xre/MbcA/ParS-like toxin binding domain (Fig. 1A). Based on their biological function described below, we renamed these genes *natR* (NAD+ toxin repressor) and *natT* (NAD+ degrading toxin). RES domain proteins (COG5654) are widespread in bacteria, including important human pathogens like *Mycobacterium tuberculosis*, *Yersinia pestis*, *Brucella abortus*, *Legionella pneumophila*, *Bordetella pertussis* or *Burkholderia pseudomallei* (Freire et al, 2019; Makarova et al, 2009; Piscotta et al, 2019). Importantly, RES toxins were shown to cleave nicotinamide dinucleotide (NAD) or to use NAD to modify targets through an ADP-ribosylation reaction, indicating that RES-based TA systems act by limiting central nicotinamide-based cofactors or by allosterically modifying downstream targets (Piscotta et al, 2019; Freire et al, 2019). Recent studies have also implicated NADase toxins in bacterial defense against bacteriophages through a self-destructive process called abortive infection (Koopal et al, 2022; Garb et al, 2022; Zaremba et al, 2022; Ofir et al, 2021). However, known phage defense systems targeting NAD either engage TIR (Toll/IL-1 receptor) or SIR2 (sirtuin) NADases, ancient immune

[1]Biozentrum, University of Basel, Basel, Switzerland. [2]Institute of Molecular Systems Biology, ETH Zürich, Zürich, Switzerland. [3]Present address: Department for Biosystems Science and Engineering, ETH Zürich, Basel, Switzerland. [4]These authors contributed equally: Isabella Santi, Raphael Dias Teixeira. ✉E-mail: urs.jenal@unibas.ch

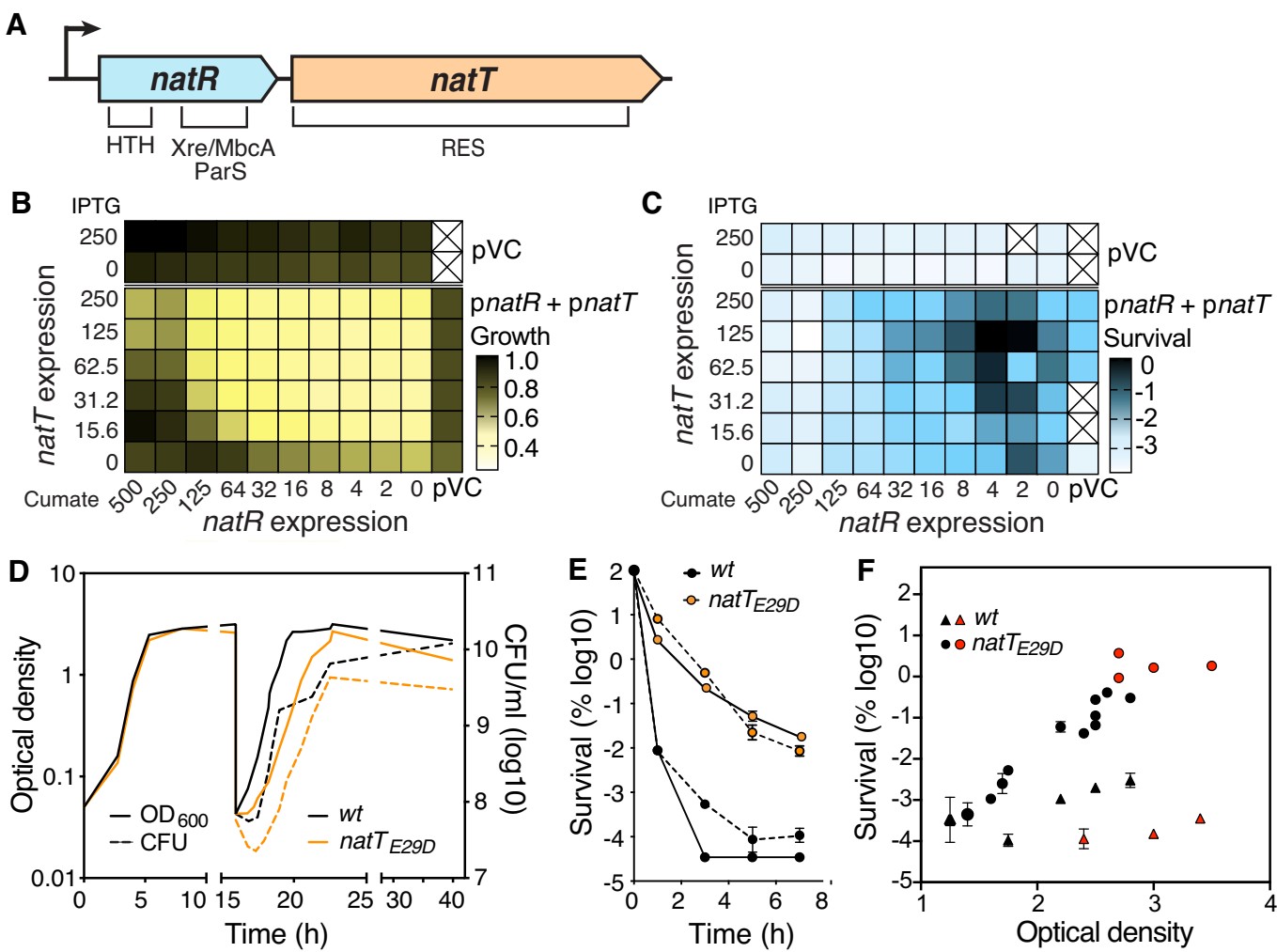

**Figure 1. The NatT toxin confers drug tolerance to *P. aeruginosa*.**

(A) Schematic of the *natR–natT* locus and domain structure of the NatT toxin and NatR anti-toxin. (B, C) Expression of *natT* limits *P. aeruginosa* growth and increases tolerance. A Δ*natR*-Δ*natT* mutant carrying plasmids with an IPTG-inducible copy of *natT* and a cumate-inducible copy of *natR* was grown in LB supplemented with different concentrations of IPTG and cumate, as indicated. Growth (B) and tolerance to ciprofloxacin (2.5 µg/ml) (C) were recorded for each combination. Heatmaps in (B) show ODs reached in the presence of the inducers normalized to the highest OD value of the analyzed group. Heatmaps in (C) show the fraction of surviving cells after treatment with ciprofloxacin (pVC control plasmid). (D) Growth of *P. aeruginosa* wild-type and *natT*$_{E29D}$ mutant in LB complex media. Stationary phase cultures were diluted into fresh medium at 15 h post inoculation. (E) Survival of *P. aeruginosa* wild-type and *natT*$_{E29D}$ mutant upon exposure to tobramycin (16 µg/ml, solid lines) and ciprofloxacin (2.5 µg/ml, stippled lines) (average ± SEM, $n = 3$). (F) Tolerance is growth phase-dependent. Cultures of *P. aeruginosa* wild-type (triangles) and *natT*$_{E29D}$ mutant (circles) were harvested at different stages of growth (indicated by OD) and exposed to tobramycin (16 µg/ml) for 3 h. Fractions of surviving cells from overnight stationary phase cultures are indicated in red (average ± SEM, $n = 3$). Source data are available online for this figure.

modules that have spread from bacteria to animals and plants, where they have adopted important roles in innate immune defenses (Wein and Sorek, 2022).

Most RES domains are part of bacterial TA systems (Makarova et al, 2009). Although TA systems have been shown to play prominent roles in abortive infection and phage defense, none of the RES domain TA systems has been implicated in bacterial immunity, leaving the physiological role of this large protein family unresolved. Some members have recently been implicated in solvent tolerance, plasmid stability or stress-mediated growth inhibition (Skjerning et al, 2019; Piscotta et al, 2019; Freire et al, 2019; Takashima et al, 2022; Kusumawardhani et al, 2020). For example, the MbcA-MbcT system of *M. tuberculosis* is significantly upregulated in a variety of stress conditions, including persister cells (Keren et al, 2011), hypoxic stress (Rustad et al, 2008), starvation (Gupta et al, 2017), and in human macrophages (Homolka et al, 2010; Ariyachaokun et al, 2020). Similarly, the *natR–natT* module of *P. aeruginosa* is induced under oxidative stress conditions, exposure to antibiotics and under host-like conditions (Teitzel et al, 2006; Zadeh et al, 2022; Song et al, 2023) and *natT* mutants show reduced survival during antibiotic treatment and in macrophages (Song et al, 2023). Over-expression of *natT* alone, but not co-expression of *natR* and *natT*, caused a growth arrest in *E. coli*, indicating that NatR–NatT operates as a bona fide TA system (Song et al, 2023). While this indicated that RES domain TA systems are involved in cellular homeostasis and stress response, the mechanisms controlling their activity remain unknown.

Here, we show that an active variant of NatT confers strongly increased tolerance to *P. aeruginosa*. We show that NatT is an NAD phosphorylase, which leads to the depletion of both NAD and NADP in a subpopulation of cells. Importantly, while actively growing *P. aeruginosa* cells overcome toxin-mediated NAD deficiencies by inducing the NAD salvage pathway, NatT generates dormant, NAD-depleted cells under nutrient-limited conditions, which spawn hyper-tolerant persisters during outgrowth. Supplementing the growth medium with the NAD precursor nicotinamide blocks toxin expression and activation and eliminates persister formation. Structure-function analyses of the NatR–NatT complex show how NatT interacts with its cognate partner NatR, indicating how this interplay leads to catalytic activation and autoregulation of the NatT toxin. These studies identify a TA system in *P. aeruginosa* that can drive persister formation by modulating a key metabolite of energy metabolism.

## Results

### The NatT toxin induces persister formation in *P. aeruginosa*

We first deleted the *natR and natT* genes on the *P. aeruginosa* chromosome and found that this did not affect growth or fitness under laboratory conditions and did not change survival in the presence of different classes of antibiotics (Fig. EV1A–C). In contrast, ectopic expression of *natT* from a plasmid increased *P. aeruginosa* survival during drug treatment without affecting growth rates (Fig. EV1D–F). Tuning *natR* and *natT* expression individually from different inducible promoters, gradually compromised growth and boosted drug tolerance, with increased NatT levels being countered by increased NatR (Fig. 1B,C). Ectopic expression of $natT_{E29D}$, a mutant isolated in a screen for increased tolerance in *P. aeruginosa* (Santi et al, 2021), showed stronger interference with growth (Fig. EV1D) and strongly increased survival during exposure to tobramycin (Fig. EV1E) or ciprofloxacin (Fig. EV1F). Thus, the E29D mutation boosts toxin-associated phenotypes, suggesting that it unleashes NatT activity.

To investigate how NatT influences *P. aeruginosa* growth and physiology, we replaced the chromosomal *natT* copy with $natT_{E29D}$. The resulting mutant showed normal growth in complex media, but reduced overall fitness and survival in the stationary phase (Figs. 1D and EV1A). Diluting stationary phase cultures into fresh media containing tobramycin or ciprofloxacin, showed up to 10,000-fold increased survival compared to wild-type (Fig. 1E), while resistance levels were unaltered (tobramycin: 1 μg/ml (wt); 1.5 μg/ml ($natT_{E29D}$); ciprofloxacin: 0.125 μg/ml (wt); 0.0625 μg/ml ($natT_{E29D}$)). Increased expression of $natT_{E29D}$ gradually lowered growth rates, whereas survival rates reached maximal levels already at expression strengths that did not compromise growth (Fig. EV1G,H). NatT-mediated tolerance was low in rapidly growing cells but strongly increased when cells entered the stationary phase (Fig. 1F), indicating that NatT-mediated tolerance is coupled to nutrient limitations.

### NatT is an NAD⁺/NADP⁺ phosphorylase

While expression of NatT in *E. coli* failed to yield soluble protein, combined expression with NatR produced a soluble but inactive

NatR–NatT complex. We thus purified the toxin directly from *P. aeruginosa* by affinity chromatography using strains expressing functional FLAG-tagged versions of NatT. Both NatT and $NatT_{E29D}$ were copurified with NatR (Fig. EV2A), but only the complex containing $NatT_{E29D}$ showed activity (Fig. EV2B,C). Purified NatR–$NatT_{E29D}$ rapidly degraded NAD⁺ generating ADP-ribose-1"-phosphate (ADPR-1P) and nicotinamide (NAM) as final products (Fig. 2A,B). In accordance with the production of ADPR-1P, NAD⁺ degradation was only observed in the presence of phosphate (Fig. EV2C), demonstrating that NatT is a NAD⁺-dependent phosphorylase. Likewise, purified NatR–$NatT_{E29D}$ was able to degrade NADP⁺ (Fig. EV2D).

To investigate if NatT depletes NAD⁺ and NADP⁺ in vivo, concentrations of NAD⁺/NADH and NADP⁺/NADPH were determined in cultures of *P. aeruginosa* containing plasmid-driven copies of *natT* or $natT_{E29D}$ harvested in exponential or stationary phase. Both dinucleotides were strongly reduced in cultures expressing $natT_{E29D}$ compared to wild-type (Figs. 2C and EV2E). Expression of $natT_{E29D}$ also reduced cellular pools of glutathione and succinate, while intermediates of the pentose-phosphate pathway were increased (Fig. 2D). Thus, NatT toxin activity appears to cause redox imbalances, which in turn trigger compensatory metabolic reactions. NAD⁺ and NADP⁺ depletion required NatT catalytic activity, as expression of $NatT_{E29D\ R82A}$, lacking one of the catalytic core residues of RES domains (see below), did not alter cellular levels of NAD⁺ or NADP⁺ (Figs. 2c and EV2E), did not affect growth (Fig. EV2F) and failed to confer drug tolerance (Fig. 2E). From this, we concluded that NatT is a NAD⁺/NADP⁺ phosphorylase, which leads to the depletion of both cofactors and to dynamic changes in *P. aeruginosa* metabolism.

### Interaction of NatR and NatT mediates toxin activity

To elucidate the molecular details of NatT activity and its control by NatR, we solved the crystal structure of the NatR–NatT complex. NatR–NatT adopts a hexameric structure with two central NatT molecules flanked by two dimers of NatR (NatR' and NatR") (Fig. 3A). Consistent with the hexameric structure, SEC-MALS analysis revealed a constant mass of 112 kDa in solution (Fig. EV3A). NatT is composed of a peripheral ring of α-helices and two central β-sheets that form a cavity (Fig. EV3B). Although NatT is structurally similar to other RES domain proteins (Skjerning et al, 2019; Piscotta et al, 2019; Freire et al, 2019) it harbors an additional helix–loop–helix domain (hereafter referred to as 'Flap'), which provides most of the interaction surface between NatT and NatR' (Figs. 3B,C and EV3B–D). The Flap contains several conserved negative charges (E26, D38, E42, and E47), which form electrostatic interactions with positive residues in helices α1, α4, and α7 of NatR' (Fig. 3C,D). For example, E29 of NatT forms an electrostatic interaction with R119, the last but two amino acids of the C-terminal helix α7 of NatR' (Fig. 3E). Intriguingly, the Flap region was inserted at the same position of the RES domain fold multiple times independently during evolution (Fig. EV3D; Appendix Fig. S1), indicating that it has adopted a key role in RES toxin control by modulating toxin–antitoxin interaction. In line with this idea, the activating mutation E29D is located precisely at the hinge between the NatT Flap and helix α7 of NatR', which controls access to the catalytic cavity of

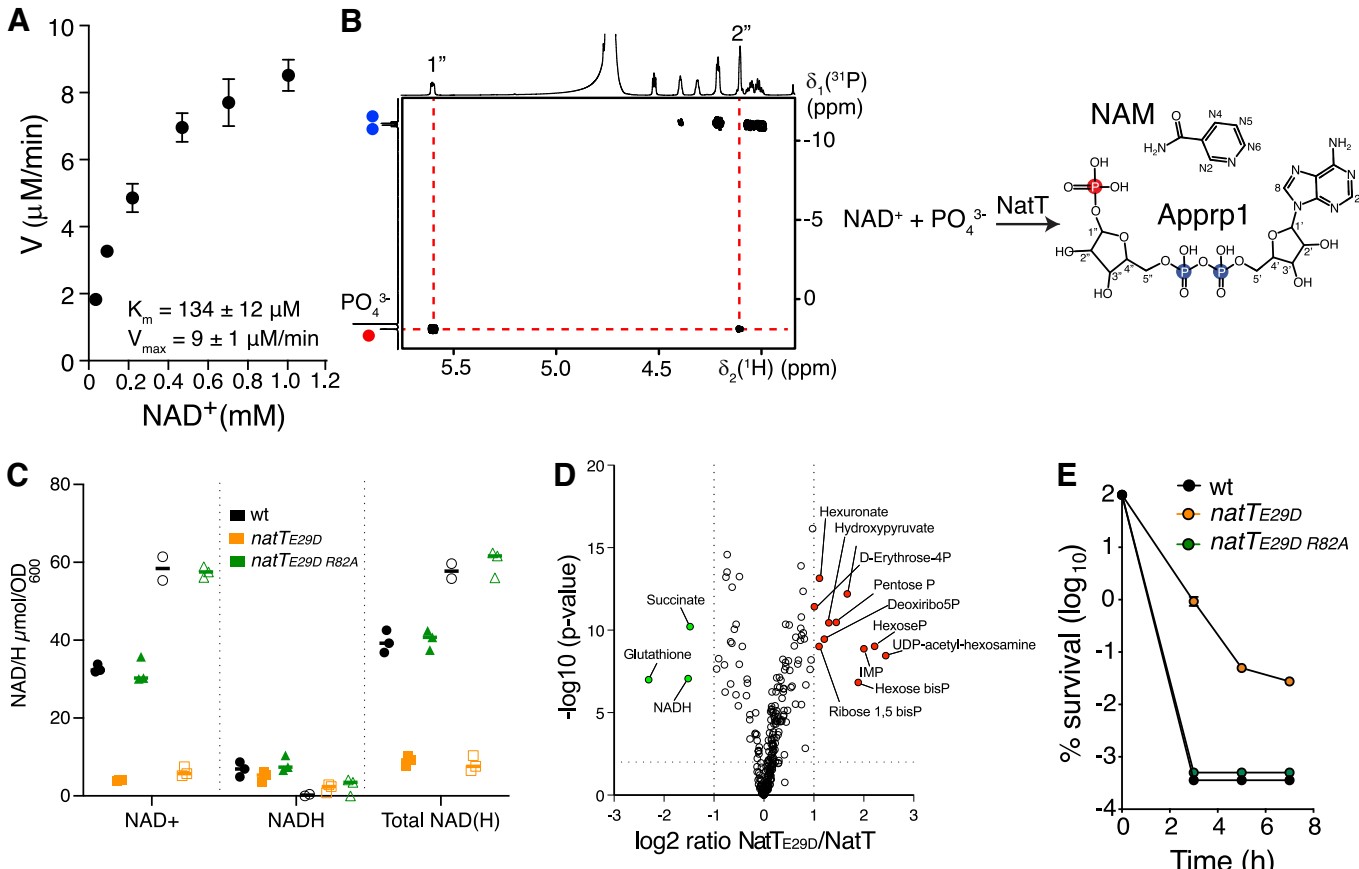

**Figure 2. NatT is a NAD-dependent phosphorylase.**

(A) Kinetics of NAD$^+$ phosphorolysis using purified NatR–NatT$_{E29D}$ complex (50 nM). The kinetic of the reaction was determined by quantifying the NAD$^+$ concentration over time in a series of 1D $^1$H NMR spectra. Km and Vmax values were determined by nonlinear regression analysis with the Michaelis–Menten equation (average ± SEM, n = 3). (B) 2D $^1$H$^{31}$P HMBC NMR NMR spectrum identifying ADPR-1P as the reaction product of NAD$^+$ (5 mM) with 40 nM purified NatRT$_{E29D}$ complex. Phosphate atoms from the ADP-ribose moiety are marked in blue and the phosphate derived from an added orthophosphate group is in red. (C) NatT depletes cellular NAD$^+$ pools of *P. aeruginosa*. NAD$^+$ and NADH concentrations were determined in *P. aeruginosa* wild-type (wt) and in strains harboring plasmid-encoded copies of *natT*$_{E29D}$ and *natT*$_{E29D\ R82A}$ growing exponentially (filled bars) or in stationary phase (open bars) (technical replicates, average ± SEM, n ≥ 2). (D) Expression of *natT*$_{E29D}$ induces global changes of *P. aeruginosa* metabolites. Comparison of metabolites in strains harboring a plasmid with an IPTG-inducible copy of *natT*$_{E29D}$ or a control plasmid. Metabolites significantly enriched or depleted in the *natT*$_{E29D}$ strain are shown in red and green, respectively (n = 3). The P value was obtained from a t test of the metabolite levels. (E) NatT toxin activity is required for *P. aeruginosa* drug tolerance. Survival of different *P. aeruginosa* strains is shown during exposure to tobramycin (16 μg/ml) (technical replicates, average ± SEM, n > 3). Source data are available online for this figure.

NatT (Figs. 3E and EV3D). Moreover, a well-defined tetrahedral atomic density modeled as a phosphate molecule was found in the interface between the Flap and NatR', in close vicinity of residue E29 (Figs. 3C,E and EV3B). Residues coordinating the phosphate moiety (R70 and R119 of NatR'; K54, Y66, E104 and Y74 of NatT) are highly conserved in NatR and NatT homologs and form a pocket at the NatT-NatR' interface (Figs. 3E and EV3D).

Superimposing the structures of NatT and diphtheria toxin bound to its NAD$^+$ substrate indicated that the active site of NatT is located at the base of the central cavity formed by its β-sheet core (Bell and Eisenberg, 1996). Active site residues are conserved in NatT and other RES domain proteins, including R23, R82, Y93, N193, and S184 (Fig. EV3B,D) (Freire et al, 2019). Residues R82, Y108, Y162, and R186 coordinate a second phosphate molecule in the active site of NatT, positioned next to the nicotinamide group of NAD$^+$, that likely participates directly in the cleavage of NAD$^+$

(Fig. EV3B). In the NatR–NatT complex, the C-terminal helix α7 of NatR' extends all the way into the substrate-binding cavity of NatT (Fig. 3B,C), indicating that NatR' controls toxin activity by blocking substrate access and that dynamic repositioning of α7 and NatR' is likely required to activate NatT.

To explore the NatT activation mechanism, we next solved the structure of NatT$_{E29D}$ in complex with NatR. Structural differences between wild-type and mutant proteins were limited to subtle changes in the NatR'-NatT interface close to residue E29. A water molecule bridging the phosphate positioned in the NatR'-NatT interface with residues T70 and R73 of NatT wild-type was missing in the E29D mutant (Fig. 3E). Moreover, the E29D substitution changes its interaction with R119 of NatR', forcing the neighboring E26 to adopt a different rotameric configuration, which in turn disrupts the interaction with the main chain of the C-terminal helix α7 of NatR'. Based on this, we speculate that the E29D mutation

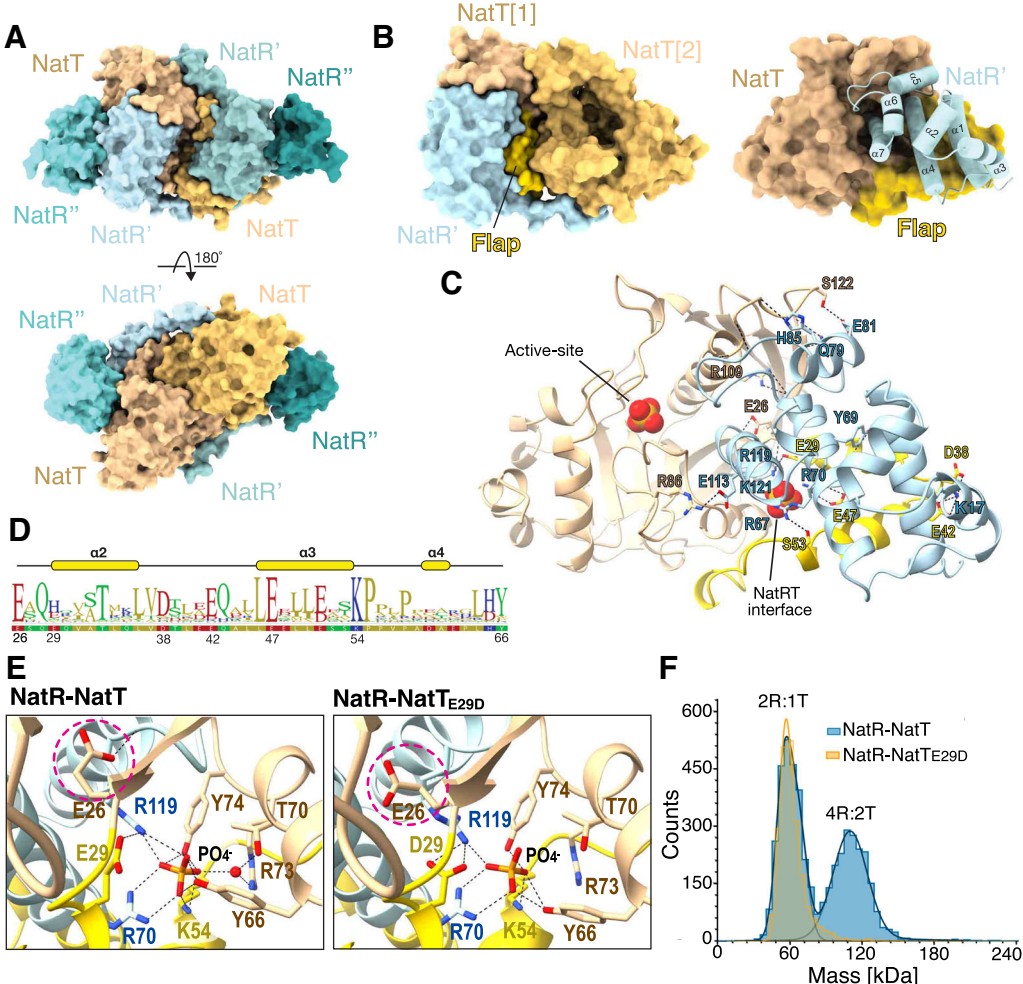

**Figure 3. Interaction of NatR and NatT mediates toxin activation.**

(A) Crystal structure of a NatR–NatT hexamer with a central NatT dimer and two flanking dimers of NatR (NatR' and NatR"). (B) NatR–NatT interaction is mediated by the Flap region. Left: Surface representation of NatT [protomer 1] with its Flap region (yellow) sandwiched between the interacting NatR' and the neighboring NatT [protomer 2]. Right: Surface representation of NatT with NatR in cartoon style and individual helices marked. NatR' interacts with the Flap region of NatT and α7 of NatR' extends into the active site groove of NatT. (C) Detailed view of NatT-NatR' interface with negatively charged residues of the Flap (yellow) and positively charged residues of NatR' indicated in stick representation. Phosphate moieties in the active site and in the NatR–NatT interface are indicated as space-filled molecules. (D) Conservation of Flap region (NatT residue 26–66). Positions of conserved residues interacting with NatR' are highlighted. K54 and Y66 coordinate the phosphate molecule in the NatR–NatT interface (see: E). (E) Zoom-in views comparing the interaction of NatR' with NatT wild-type and NatT$_{E29D}$. Residues with altered conformation are highlighted in stick representation with coloring scheme as in (C). The movement of residue E26 away from its interaction with the backbone of α7 of NatR' (blue) is marked by a red circle. (F) Mass photometry analysis of NatR–NatT (blue) and NatR–NatT$_{E29D}$ (yellow) at 10 nM. The mass of the NatR–NatT$_{E29D}$ complex corresponds to a NatR'-NatR"-NatT complex. Source data are available online for this figure.

weakens the interaction between the NatT Flap and NatR', thereby promoting the disassembly of the stable hexameric structure. This is in line with SEC-MALS experiments showing that the NatT-NatR wild-type complex is a stable hexamer (112 kDa) at micromolar concentrations, whereas the complex containing NatT$_{E29D}$ displayed an additional peak with significantly lower mass (Fig. EV3A). Mass photometry analysis of the NatR–NatT wild-type complex at low nanomolar protein concentrations revealed two different oligomeric states, a 112 kDa hexamer and a 60 kDa species, which likely corresponds to a complex between a NatR dimer and NatT (2 R:1 T) (Fig. 3F). Importantly, the NatR–NatT$_{E29D}$ complex showed a single peak at 60 kDa, indicating that the TA complex is in a dynamic equilibrium between an inert hexameric state and a

smaller, active subcomplex and that this transition mediates NatT toxin activation. Of note, the Flap is tightly sandwiched between NatR' and the adjacent NatT protomer (Fig. 3B), restricting Flap movement and keeping the active site pocket occluded by the C-terminus of NatR'. Disruption of the NatT-NatT dimer could liberate the Flap, allowing it to flexibly move while still bound to NatR', thereby providing access to the active site.

## NatT activity is coupled to *natR–natT* transcription

When monitoring NatR and NatT protein levels in different *P. aeruginosa* strains, we noticed that both proteins are elevated in the *natT*$_{E29D}$ mutant compared to wild-type (Fig. 4A,B). To examine if

this results from altered transcription, we engineered *P. aeruginosa* strains carrying a *gfp* reporter downstream of *natR–natT* on the chromosome (Figs. 4A and EV4A). Expression of *natT* was strongly upregulated in a Δ*natR* mutant, but reduced to wild-type levels when *natR* was complemented from a plasmid (Figs. 4C and EV4B). Likewise, in a Δ*natT* mutant transcription of *natR–natT* was derepressed (Fig. 4C) and NatR levels strongly upregulated (Fig. 4D). Thus, *natR–natT* transcription is autoregulated with

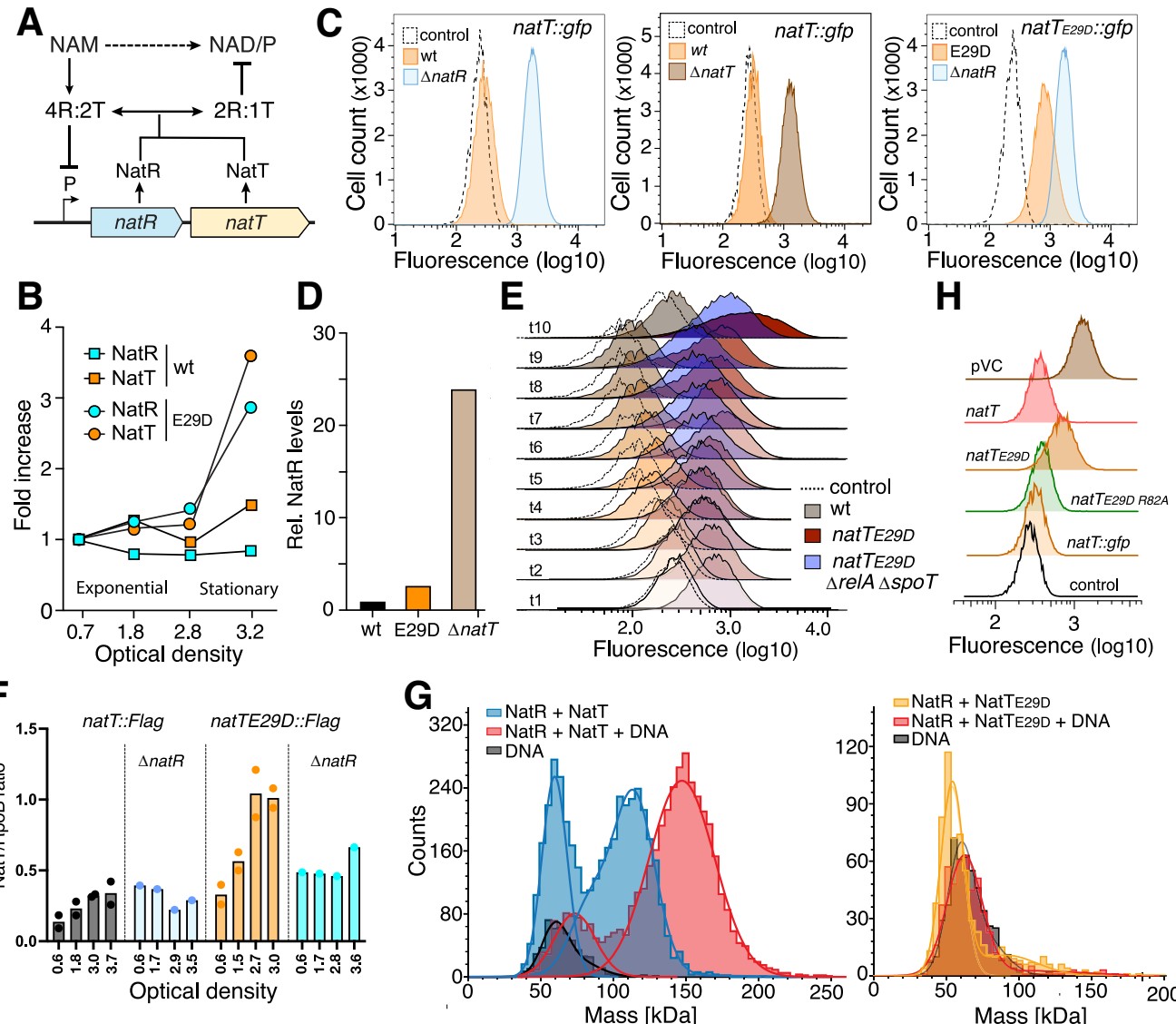

**Figure 4. NatR is an anti- and a co-toxin of NatT.**

(A) Schematic model of NatT activation and autoregulation. A catalytically inactive hexameric complex (4R:2T) acts as repressor of *natR–natT* transcription. Transition to a 2R:1T complex leads to derepression of *natR–natT* transcription and under specific conditions to NatT NADase activation. The postulated regulatory role of the NAD precursor nicotinamide (NAM) is indicated. (B) NatR (turquoise) and NatT (orange) levels are increased in a *natT_{E29D}* mutant. Proteins were quantified by mass spectrometry and plotted as a function of growth (optical density) of *P. aeruginosa* wild-type (wt) and *natT_{E29D}* mutant. (C) Transcription of *natT* is controlled by the NatR-NatT complex. Transcription was determined in reporter strains containing a chromosomal copy of *gfp* downstream of *natT*. Fluorescence of a control strain lacking the *gfp* reporter is indicated (dotted lines). (D) NatR protein levels are increased in *natT_{E29D}* and Δ*natT* mutants. Levels of NatR were determined by quantitative mass spectrometry analysis of *P. aeruginosa* wild-type and mutants indicated. (E) Transcription of *natR–natT* is induced in the *natT_{E29D}* background. Cultures of *P. aeruginosa* wild-type, *natT_{E29D}* and *natT_{E29D}* Δ*relA* Δ*spoT* mutants harboring a chromosomal *natT::gfp* reporter were assayed by flow cytometry during different phases of growth (see: Fig. EV4C). Black lines show baseline auto-fluorescence of a strain lacking a *gfp* reporter gene. (F) NatT protein levels are reduced in a strain lacking NatR. Immunoblots of *P. aeruginosa* wild-type and Δ*natR* mutants with chromosomal *natT*-FLAG or *natT_{E29D}*-FLAG alleles. Cell extracts were harvested at different ODs and stained with anti-FLAG and anti-RpoB antibodies and NatT levels normalized to RpoB. (G) NatR-NatT binds to its own promoter. Mass photometry of NatR-NatT (blue) and NatR-NatT_{E29D} (yellow) without and with DNA (red) containing the *natR* promoter region. The mass of dsDNA is shown in black. (H) Transcription of *natR–natT* activity is coupled to NatT activity. Expression of a chromosomal *natR-ΔnatT::gfp* reporter was assayed by flow cytometry in a *P. aeruginosa* Δ*natT* mutant expressing different *natT* alleles from a plasmid, as indicated. The black line represents the fluorescent signal in a control strain lacking a *gfp* reporter. One representative experiment is shown (pVC = plasmid control). Source data are available online for this figure.

both NatR and NatT contributing to its repression. Intriguingly, while *natR–natT* transcription remained at low levels in *P. aeruginosa* wild-type, it strongly increased in the $natT_{E29D}$ background as cells progressed toward stationary phase (Figs. 4E and EV4C). NatT and NatR protein levels were also elevated in the $natT_{E29D}$ background, with differences being most prominent in the stationary phase (Figs. 4F and EV4D,E).

The above findings indicated that *natR–natT* transcription is coupled to the activity of the NatT toxin. In agreement with this, purified NatR–NatT wild-type complex (4R:2T) readily bound to the *natR–natT* promoter region, while a purified $NatT_{E29D}$ complex failed to bind the same DNA fragment (Fig. 4G). Moreover, while *natR–natT* transcription was derepressed in the $natT_{E29D}$ background, it remained fully repressed in a $natT_{E29D\ R82A}$ mutant, encoding a catalytically inactive NatT toxin (Fig. 4H). Together with the observation that $NatT_{E29D}$ failed to form a stable hexameric complex with NatR in vitro (Fig. 3F), these findings indicated that NatT toxin activity and *natR–natT* expression are controlled through a dynamic equilibrium between a stable NatR–NatT hexamer and a smaller NatR–NatT complex that is unable to adopt the conformation needed to bind to the *natR–natT* promoter region (Fig. 4A). This model is supported by the observation that increasing cellular levels of NatT not only increased auto-toxicity and drug tolerance (Fig. 1B,C) but also led to the progressive activation of the *natR–natT* promoter (Fig. EV4F), presumably by shifting the NatR–NatT equilibrium and hexameric repressor conformation.

Despite of *natT* transcription being derepressed in strains lacking NatR, NatT activity was completely abolished in a Δ*natR* mutant (Fig. 1B,C). This indicated that NatR acts as anti-toxin blocking NatT expression and activity, and at the same time, is required for toxin function. In line with this, NatT protein levels remained low in a Δ*natR* mutant (Fig. 4F), most likely due to its poor solubility and rapid degradation in the absence of NatR (Fig. EV4G,H). Deleting *natR* abolished NatT toxin activity (Fig. EV4I) as well as NatT-mediated drug tolerance (Fig. EV4J–L). From this, we concluded that NatR serves as anti- and co-toxin for NatT and that toxin regulation is mediated by subtle changes in NatR–NatT interaction and stoichiometry, a process that strictly requires NatR.

## NatT-mediated antibiotic tolerance is countered by the NAD salvage pathway

*P. aeruginosa* replenishes its $NAD^+$ pool via de novo synthesis from aspartate or, if available, from nicotinamide (NAM) through the $NAD^+$ salvage pathway I (NSPI) (Fig. 5A) (Okon et al, 2017). Intriguingly, the NAD salvage pathway is essential for *P. aeruginosa* growth under conditions mimicking human infections (Belanger et al, 2022). Transcription of the salvage pathway genes is controlled by NrtR, a repressor that binds ADP-ribose, one of the breakdown products of NAD+, leading to salvage pathway derepression (Okon et al, 2017). Deletion of NSPI genes *pncB*, *nadD2*, and *nadE* strongly increased NatT-mediated toxicity (Fig. 5A), indicating that the $NAD^+$ salvage pathway balances $NAD^+$ shortages in cells inducing NatT toxin activity. To test this, we engineered a strain carrying a *gfp* reporter downstream of *pncB1* on the *P. aeruginosa* chromosome and showed that deletion of

$nrtR^{31}$ led to strong derepression of *pncB1* (Fig. 5A,B). Moreover, *pncB1* was derepressed in a small subpopulation of the $natT_{E29D}$ mutant but not of *P. aeruginosa* wild-type (Fig. 5B,C). Cells inducing NSPI genes were not observed in a $natT_{E29D\ R82A}$ background, demonstrating that derepression of the $NAD^+$ salvage pathway requires an active NatT toxin (Fig. 5C). Increasing $natT_{E29D}$ expression led to an increased fraction of cells with derepressed NSPI genes, while boosting *natR* expression had the opposite effect (Fig. EV5A–C). Time-lapse microscopy of the $natT_{E29D}$ mutant harboring the *pncB1-gfp* reporter revealed that cells stochastically inducing the $NAD^+$ salvage pathway transiently slowed growth but quickly resumed concomitant with the loss of the fluorescent signal (Movie EV1). We also observed few cells with a strong fluorescence signal that remained high throughout the time-lapse experiment, which was not able to resume growth (Movie EV2). Thus, under nutrient-rich conditions, derepression of the $NAD^+$ salvage pathway seems to effectively neutralize stochastic NatT activation. Intriguingly, the fraction of cells inducing NSPI genes gradually declined during late log phase and upon entry into the stationary phase, indicating that the metabolic response of *P. aeruginosa* to NatT activity collapses as nutrients become limited (Fig. 5C).

To further scrutinize this idea, we supplemented the growth medium with NAM, the precursor of the $NAD^+$ salvage pathway (Fig. 5A). NAM not only reduced NatT-mediated drug tolerance in a concentration-dependent manner (Fig. 5D), but also abolished NatT-mediated derepression of the $NAD^+$ salvage pathway (Fig. 5E), restored growth of strains expressing $natT_{E29D}$ from a plasmid (Fig. EV5D), alleviated NatT-mediated hypersensitivity of an $NAD^+$ salvage pathway mutant (Fig. EV5E), and reduced the fraction of growth-arrested cells during outgrowth from stationary phase (Fig. EV5F). NAM supplementation also effectively blocked *natR–natT* transcription in the $natT_{E29D}$ mutant (Fig. 5F) and reduced levels of NatR and NatT proteins in the same strain (Fig. EV5G). Finally, genetic derepression of salvage pathway genes in an Δ*nrtR* mutant greatly reduced NatT-mediated persister formation (Fig. 5A,G).

The observation that NAM neutralized the physiological consequences of NatT even in the absence of a functional salvage pathway (Fig. EV5E), indicated that it abolished persister formation by interfering with toxin expression and/or activity rather than by simply replenishing pools of nicotinamide dinucleotides. A possible candidate mediating NAM-specific effects is the global alarmone (p)ppGpp, which controls bacterial physiology under nutrient-deplete conditions (Hauryliuk et al, 2015) and was implicated in antibiotic tolerance (Stewart et al, 2015; Pacios et al, 2020; Martins et al, 2018). To test this, we introduced the $natT_{E29D}$ allele into a strain lacking RelA and SpoT, the enzymes responsible for (p)ppGpp synthesis (Boes et al, 2008). While NatT-mediated tolerance was indeed completely abolished in a strain lacking (p)ppGpp (Fig. EV5H,I), this effect was not due to interference with NatT activity as both *natR–natT* transcription (Fig. 4E) and derepression of the $NAD^+$ salvage pathway (Fig. 5C), were unchanged in the Δ*relA* Δ*spoT* mutant. Thus, (p)ppGpp does not promote *P. aeruginosa* drug tolerance by regulating NatT toxin activity, but rather by controlling downstream processes that help *P. aeruginosa* cope with the physiological consequences of NatT-mediated $NAD^+$ depletion.

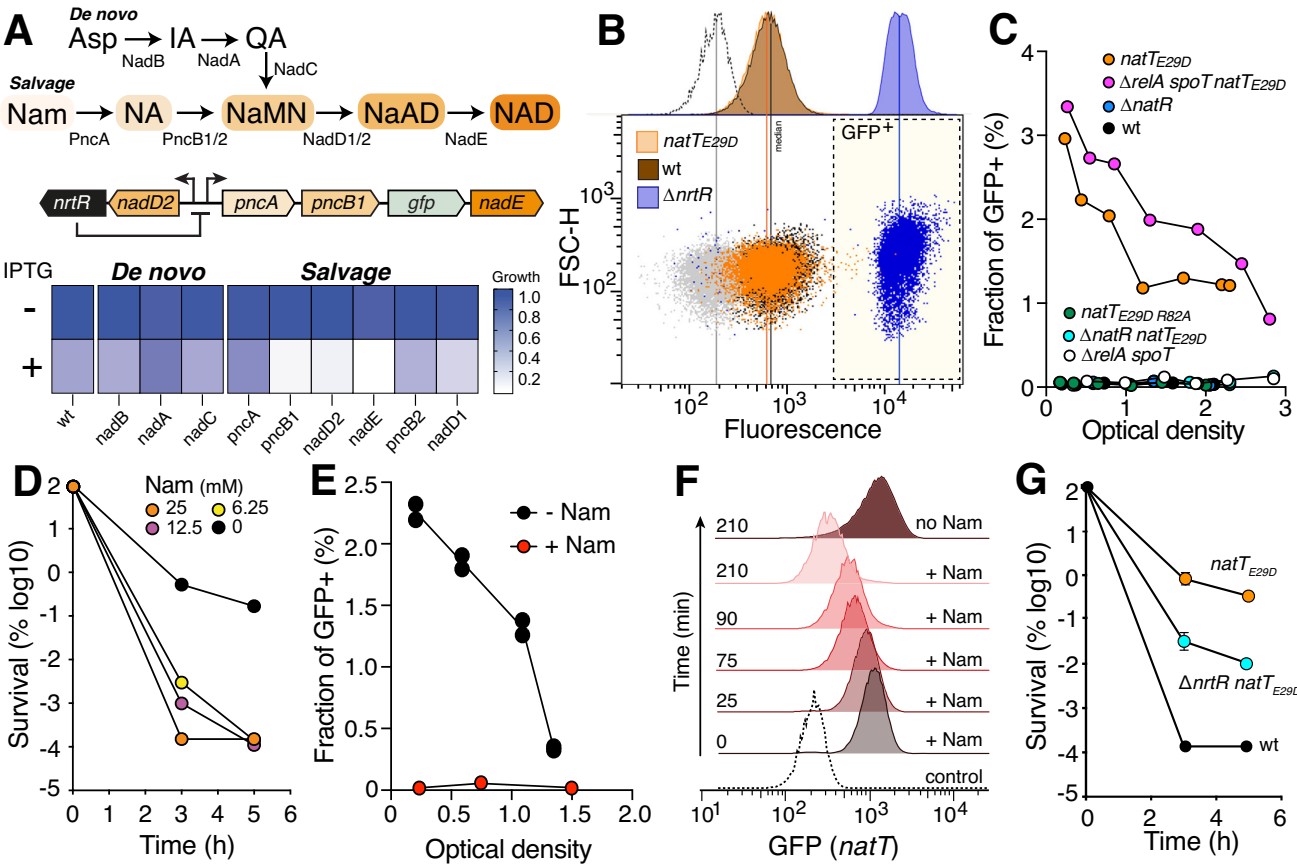

**Figure 5. The NAD salvage pathway neutralizes NatT toxin activity and abolishes drug tolerance.**

(A) Schematic of NAD⁺ de novo and salvage pathways in *P. aeruginosa* (top) with intermediates and catalyzing enzymes indicated. Salvage pathway regulation by NrtR is shown below with enzymes and genes being highlighted in matching colors. The position of the chromosomal *gfp* reporter used for these studies is in green. Impact of $natT_{E29D}$ expression on the growth of *P. aeruginosa* wild-type and different salvage pathway mutants is shown at the bottom. Strains carrying an IPTG-inducible copy of $natT_{E29D}$ on a plasmid were grown in LB medium with or without IPTG. The heatmap shows the OD after 8 h of growth normalized to the OD of wild-type. (B) Transcription of NAD salvage pathway genes in individual cells of *P. aeruginosa* wild-type, $natT_{E29D}$, and Δ*nrtR* mutants harboring a *pncB1::gfp* reporter was analyzed by flow cytometry. GFP signals are shown for individual cells and as histograms. A strain lacking the *gfp* reporter is shown as control (gray dots and stippled line). The area with average GFP signals higher than wild-type (GFP⁺) is marked in yellow. Please note the small subfraction of $natT_{E29D}$ mutant cells (orange dots) with increased expression of the *pncB1::gfp* reporter (yellow stippled box). (C) NatT-mediated activation of salvage pathway genes declines in a growth phase-dependent manner. Fractions of cells expressing salvage pathway genes are shown as a function of the optical densities of cultures analyzed. (D) Nicotinamide (NAM) abolishes NatT-mediated drug tolerance. *P. aeruginosa* $natT_{E29D}$ grown in LB supplemented with different concentrations of NAM was challenged with tobramycin (32 µg/ml) for 3 h and fractions of surviving cells were determined. (E) NAM abolishes NatT-mediated NAD⁺ salvage pathway induction. Fractions of cells with induced NAD⁺ salvage pathway were determined as in (C) in the presence or absence of NAM (20 mM) (technical replicates, average ± SEM, $n = 2$). (F) NAM blocks NatT-mediated derepression of the *natR–natT* operon. Growing cultures of the $natRT_{E29D}::gfp$ reporter strain were supplemented with NAM (25 mM) and analyzed by at times as indicated. A control strain lacking a *gfp* reporter is shown (dotted line). (G) Salvage pathway derepression partially abolishes NatT-mediated drug tolerance (technical replicates, average ± SEM, $n = 3$). Experiments were carried out as indicated in (D). Source data are available online for this figure.

## NatT generates persister subpopulations with an extended lag phase

To assess the impact of NatT on *P. aeruginosa* growth, we engineered reporter strains constitutively expressing the fluorescent protein TIMER (Claudi et al, 2014), which reads out growth rates of individual cells without compromising NatT-mediated persister formation (Fig. EV6A). While growth was not compromised in wild-type and the $natT_{E29D}$ mutant during the logarithmic growth phase, numbers of growth-arrested cells spiked up in the mutant and, to a lower extent in wild-type, upon nutrient depletion

(Figs. 6A and EV6B). In this experiment, samples from stationary phase were diluted into fresh medium and TIMER-based fluorescence was analyzed three hours after growth resumption. Spotting stationary cultures directly onto agar patches containing fresh nutrients showed that wild-type cells resumed growth immediately, while a fraction of $natT_{E29D}$ mutant cells showed lag phases of variable lengths (Fig. 6B). Deleting *nrtR*, the gene encoding the salvage pathway repressor, eliminated the fraction of $natT_{E29D}$ mutant cells with extended lag phase almost entirely (Fig. EV6C), indicating that metabolically dormant cells result from NatT-mediated NAD⁺ depletion. From this, we concluded that

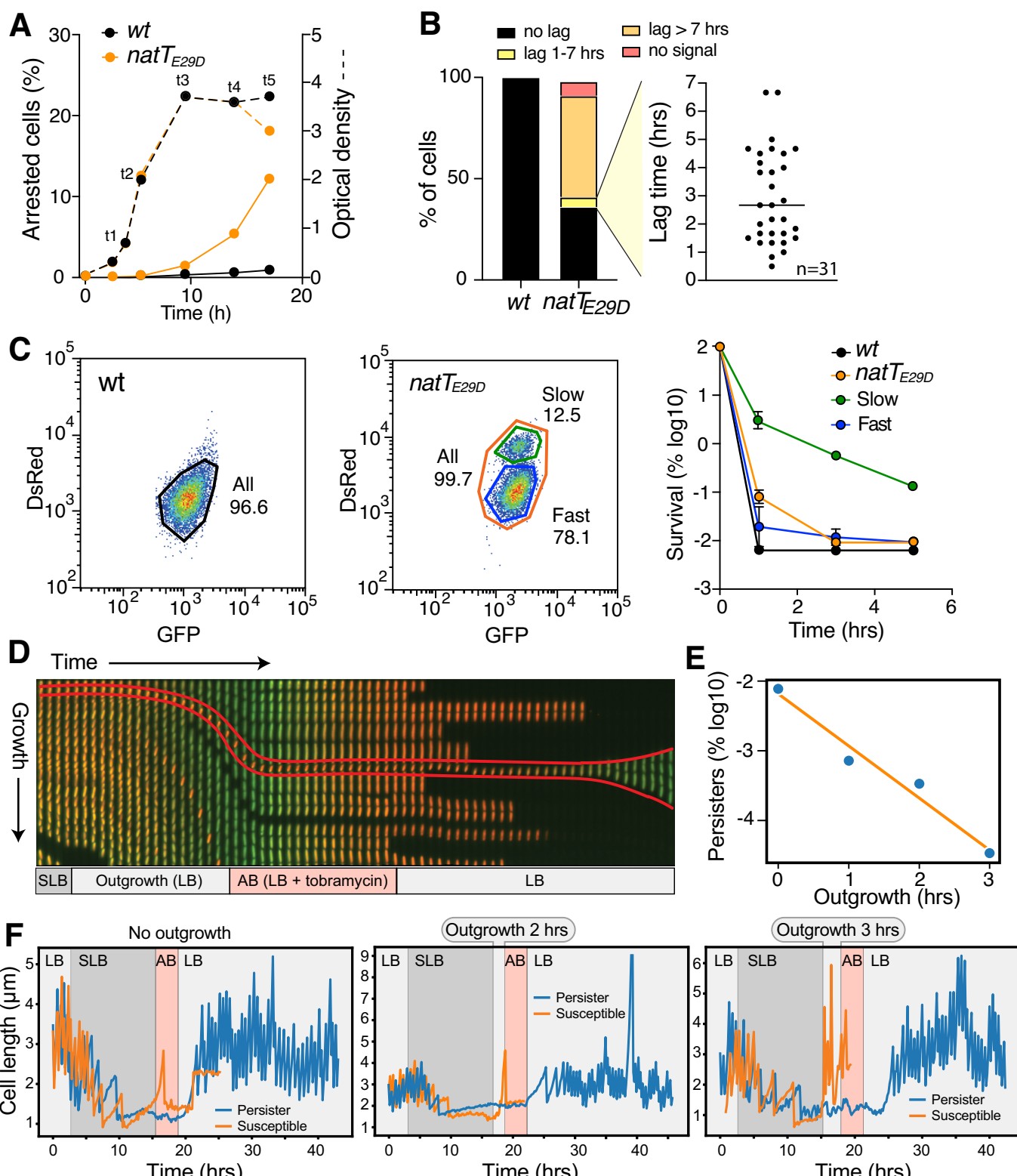

NatT activity generates arrested cells in the stationary phase that experience extended lag periods when exposed to fresh nutrients.

To investigate if the observed transient growth arrest contributes to antibiotic survival, stationary phase cultures of strains expressing TIMER were resuspended into fresh media for 3 h, followed by

FACS sorting to separate growing from arrested subpopulations using green and red fluorescence (Fig. 6C). Exposing sorted populations to high concentrations of tobramycin (16 µg/ml) revealed rapid killing of *P. aeruginosa* wild-type and the *natT_{E29D}* subpopulation that showed rapid growth recovery. In contrast, the

◄

**Figure 6. NatT activation generates persister subpopulations with extended lag phase.**

(A, B) The *P. aeruginosa* $natT_{E29D}$ mutant generates growth-arrested cells. Cultures of *P. aeruginosa* wild-type and $natT_{E29D}$ mutant expressing TIMER were grown in LB and samples were removed at indicated the time points (t1–t6), diluted into fresh medium (OD 0.1) and analyzed by flow cytometry after three hours of growth. Fractions of arrested cells are plotted in (A) (solid lines). (B) Cells harvested at time point t5 (A) were spotted on LB agar patches, and their growth was analyzed by time-lapse microscopy (wild-type, $n = 450$; $natT_{E29D}$, $n = 150$). Bar plots indicate fractions of cells with delayed re-growth and different lag times (black line=median). (C) The $natT_{E29D}$ mutant generates growth-arrested cells with increased drug tolerance. Stationary phase cultures of wild-type and $natT_{E29D}$ mutant expressing TIMER were harvested at time point t5 (A), diluted into fresh LB medium for three hours, and sorted by FACS using the red and green fluorescence channels (left panels). Subpopulations were harvested and survival was determined upon exposure to tobramycin (right panel) (technical replicates, average ± SEM, $n = 3$). (D) Dual-input mother machine with *P. aeruginosa* cells expressing TIMER growing in parallel side channels supplied by media influx from a central media channel. Fluorescence images are assembled from one single channel representing the growth of cells over time as indicated by the arrows. Cells were loaded into the microfluidic device, supplied with LB for 3 h, then gradually switched to spent LB (SLB) for 12.5 h before they were again supplied with fresh LB for 3 h (outgrowth), followed by exposure to LB containing tobramycin (16 μg/ml) for 3 h (treatment) and finally growth resumption in LB. TIMER fluorescence indicates growing (green) and resting (red) cells. (E) Persister frequency drops with increasing length of outgrowth. The frequency of cells surviving antibiotic exposure was measured in a microfluidic device as indicated in (D) with variable length of outgrowth periods. The analysis included a total of 122,704 cells with 241 persisters lineages. (F) Representative examples of persister lineages (blue) grown in microfluidic devices as indicated in (D). The growth and division of individual cells is indicated by length measurements over time. Representative susceptible lineages (orange) are shown as control. Stages during which specific media were supplied are indicated in the same color code as in (D). Source data are available online for this figure.

subpopulation with slow growth recovery showed significantly increased survival (Fig. 6C). From this, we concluded that NatT mediates metabolic dormancy during the stationary phase, which in turn protects such persisters from antibiotic killing during outgrowth in fresh media.

To establish the link between NatT activity, stationary phase, and drug tolerance at the single-cell level, we adopted a dual-input mother machine microfluidic device that enables controlled variation of growth conditions and direct observation of the response of thousands of individual cells (Kaiser et al, 2018). To mimic stationary phase physiology, *P. aeruginosa* wild-type and $natT_{E29D}$ mutant expressing TIMER were grown in microchannels and exposed to a gradual switch from complex media (LB) to spent media (SLB) harvested from the supernatant of stationary *P. aeruginosa* batch cultures (details see: "Methods"). During exposure to nutrient-limited conditions, *P. aeruginosa* executed a few rounds of reductive divisions before growth and division were stalled completely, as indicated by the gradual conversion of TIMER fluorescence from green to red. Upon infusion of fresh LB into the microchannels, the vast majority of cells rapidly resumed active growth and division, as indicated by the reversion of TIMER fluorescence from red to green (Fig. 6D). Switching channels to fresh LB containing tobramycin (16 μg/ml) resulted in rapid stalling of growth and cell killing (Fig. 6D).

While no persisters were observed when using *P. aeruginosa* wild-type, about 1% of the cells of the $natT_{E29D}$ mutant survived antibiotic treatment (Fig. 6E). Not only is this number similar to the fraction of persisters observed in liquid cultures (Figs. 1E and 5G), but survival was strictly dependent on passaging cells through SLB. Moreover, if cells were exposed to fresh media without antibiotics for increasing time periods (outgrowth) before antibiotics were added to the growth media, the number of surviving cells gradually dropped (Fig. 6D,E). Thus, NatT-mediated drug tolerance is a transient state generated during nutrient starvation and gradually lost after growth resumption.

Analyzing over 120,000 individual cells revealed a total of 241 persister lineages, indicated by their ability to survive antibiotic treatment and resume growth after drug washout (Fig. 6D). Tracing back persister lineages revealed that they are indistinguishable from susceptible cells, except that they all show prolonged SLB-induced dormancy when cells were supplied with fresh media. Three

representative examples are shown in Fig. 6F, delineating their extended stasis through outgrowth and drug treatment periods. In contrast, all susceptible lineages showed rapid growth resumption and drug-mediated killing. These experiments establish a clear model in which NatT activity in the stationary phase, directly or indirectly, generates dormant persister subpopulations.

## NatT is under positive selection during infections in humans

To assess the potential role of NatR–NatT during *P. aeruginosa* infections, we analyzed *natR* and *natT* gene sequences in 8286 *P. aeruginosa* genomes available in the SRA database (Leinonen et al, 2011). While NatR sequences are highly conserved, more than half of the strains analyzed contained single or multiple SNPs in NatT (Fig. 7A). The most frequent changes were I14V and V117A, which occurred as single SNPs or in combinations with other mutations and likely mark the PA14 strain lineage (Ozer et al, 2019). Additional *natT* SNPs were found in 67% of all isolates from CF patients ($n = 479$), 83% of the isolates from diverse acute infections ($n = 161$), and 80% of other hospital isolates ($n = 305$) (Dataset EV1), arguing that NatT toxin function is under selective pressure during human infections. The E29D mutation was not found in any of these strains, indicating that high fitness costs may prevent its selection in vivo (Fig. EV1A). A selection of the most frequent *natT* alleles was amplified from patient isolates and grafted into *P. aeruginosa* lab strain PAO1. While none of these alleles showed a toxic effect when expressed ectopically, survival rates gradually increased with increasing numbers of *natT* mutations, with one variant harboring five replacements conferring the highest level of tolerance (Fig. 7B) and activation of the NSPI salvage pathway genes similar to the $natT_{E29D}$ allele (Fig. 7C). Of note, this *natT* allele carried mutations in residues S105 and P165, which are positioned at the NatR–NatT interface, close to the active site of NatT.

These observations suggested that mutations activating NatT accumulate during infections in humans, possibly increasing stress tolerance under these conditions while balancing fitness costs. Alternatively, NatT-mediated NAD depletion and salvage pathway induction could generate a *P. aeruginosa* subpopulation with altered virulence. NrtR, the repressor of NSPI salvage pathway genes, also regulates virulence gene expression, including T3SS (Jin et al, 2019) and T6SS (Zhang et al, 2022). To test this possibility, we

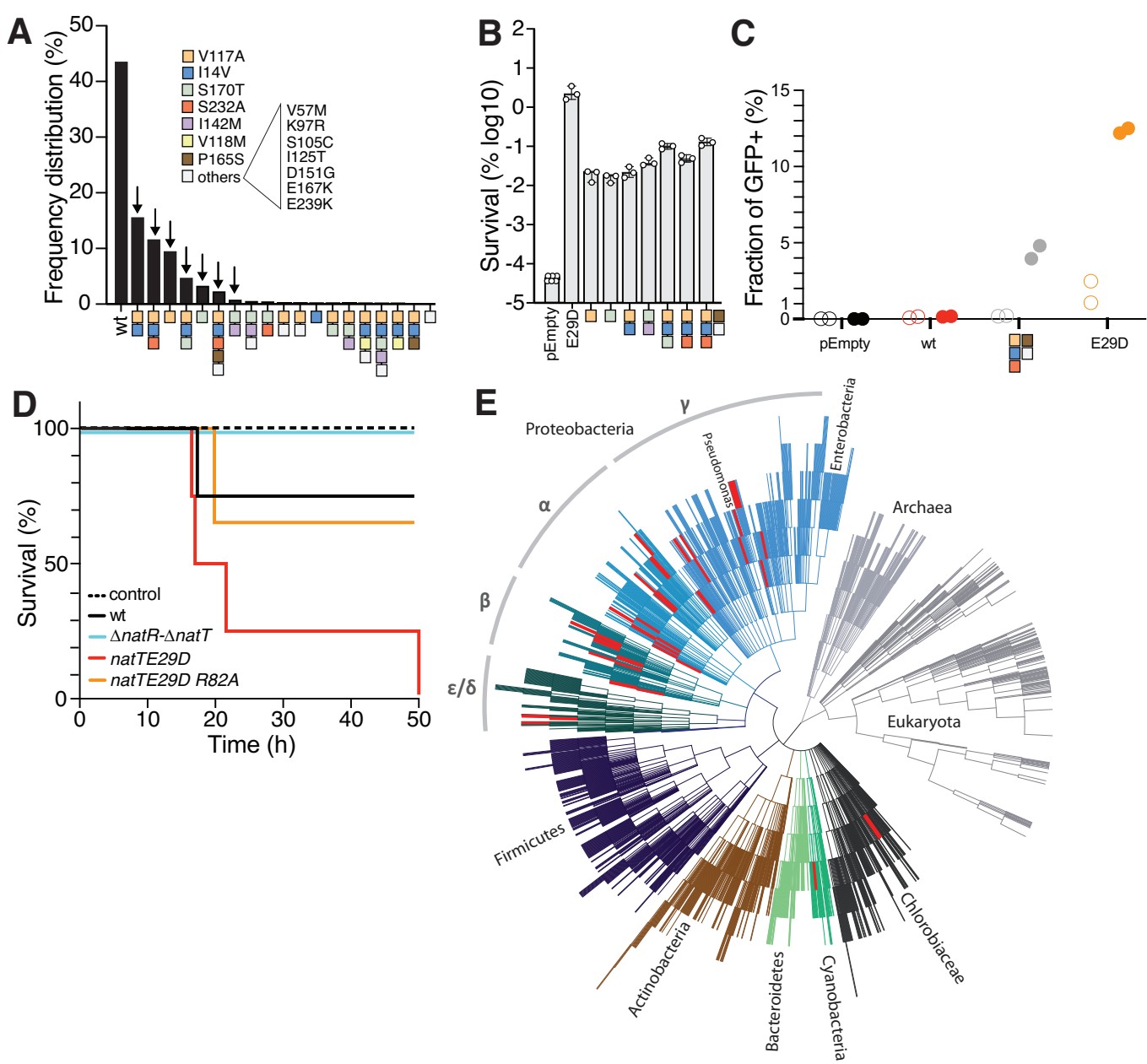

**Figure 7. The NatRT module is widespread in proteobacteria and is under selection during infections.**

(A) Frequency distribution of *natT* alleles in 8286 *P. aeruginosa* strains analyzed. Single SNPs and SNP combinations are indicated in different colors. Arrows indicate the *natT* alleles that were tested in this study. (B) Different *natT* alleles confer different levels of drug tolerance. Cultures of *P. aeruginosa* carrying plasmids with different *natT* alleles from (A) were scored for survival after 3 h of treatment with tobramycin (technical replicates, average ± SEM, n = 3). (C) Salvage pathway induction by different *natT* alleles. A *P. aeruginosa pncB1::gfp* reporter strain carrying plasmids with different *natT* alleles under IPTG control was grown with (filled bars) or without (open bars) IPTG. Fractions of cells with a derepressed salvage pathway (GFP⁺) are indicated. (D) NatT activity controls *P. aeruginosa* virulence in a simple insect larvae model. Survival rates of *G. mellonella* larvae are shown for different *P. aeruginosa* strains as indicated. Experiments were carried out with five larvae per strain tested. (E) Bacterial phylogeny with the distribution of NatR–NatT TA system orthologs (red bars). Source data are available online for this figure.

used different *natT* mutants to infect *Galleria* larvae, a simple non-vertebrate infection model for *P. aeruginosa* (Hill et al, 2014). Intriguingly, a *natT*$_{E29D}$ mutant showed strongly increased killing, while survival of larvae infected with a Δ*natT* mutant was increased (Fig. 7D). These findings are compatible with the idea that NatT toxin activation stimulates both *P. aeruginosa* virulence and persistence.

## Discussion

Self-intoxicating degradation of cellular pools of NAD/NADP is used by bacteria to defend against bacteriophages in a process called abortive infection (Koopal et al, 2022; Garb et al, 2022; Zaremba et al, 2022; Ofir et al, 2021). But while known bacterial immunity systems targeting NAD exclusively adopt TIR and SIR2

NADases (Wein and Sorek, 2022), the physiological role of RES domain toxins has remained unclear (Makarova et al, 2009; Freire et al, 2019). Here, we show that an activated variant of NatT, a RES domain toxin that is part of the core genome of *P. aeruginosa*, strongly promotes survival during treatment with different classes of bactericidal antibiotics. NatT belongs to a subgroup of RES domain proteins that have adopted a Flap region mediating its interaction with the anti-toxin NatR. Orthologs of the NatR–NatT TA system are primarily found in proteobacteria, where they seem to have spread horizontally to distinct clades in alpha-, beta-, gamma-, delta- and epsilon-purple bacteria, but are absent in enterobacteria (Fig. 7E). This wide taxonomic distribution and the observation that none of the *natR* or *natT* homologs are associated with phage genes or phage remnants, argues against NatT playing a role in phage defense. In line with this, probing a selection of more than 50 natural *P. aeruginosa* phage isolates has failed to show differential susceptibility of wild-type and a Δ*natT* mutant.

Although TA systems have been implicated in persister formation in response to stress, the idea that they contribute to the fitness of bacterial populations under adverse conditions has remained controversial (Jurėnas et al, 2022; Dörr et al, 2010, 2009; Goormaghtigh and Melderen, 2019). Stress-mediated phenotypes of TA null mutants are often missing (Verstraeten et al, 2015; Harms et al, 2017; Goormaghtigh et al, 2018; Conlon et al, 2016) and transcriptional upregulation of TA systems in response to diverse stressors did not result in active RNAse toxins in *E. coli* (LeRoux et al, 2020). In agreement with this, we found that even in strains that globally depressed *natR–natT* transcription, NatT activity remained limited to a small subpopulation, arguing that additional, post-transcriptional levels of NatT control must exist and that additional signals may be required for toxin activation. Limiting toxin activity to a small subpopulation may also explain why the stress-mediated expression of TA systems in *E. coli* does not result in observable growth inhibition (LeRoux et al, 2020). Moreover, our data showed that even when NatT is unleashed, actively growing cells can neutralize its activity by replenishing NAD. Whereas NatT-dependent derepression of NAD salvage pathway genes effectively evaded growth obstruction, salvage pathway mutants were hypersensitive to NatT. The observation that the fraction of cells with derepressed salvage pathway gradually declines during entry into stationary phase, indicated that NatT-mediated NAD/P depletion can no longer be countered when cells run out of nutrients, thereby establishing a population of persister cells. This explains why NatT-mediated persister formation requires stationary phase physiology, while actively growing cells show little drug tolerance. Thus, NatT activity together with differential regulation of salvage pathway components could adjust the formation of persisters when bacteria experience nutritional stress. If persister formation is a direct consequence of NAD/P depletion or is caused indirectly by other forms of metabolic stress in nutrient-depleted cells, remains to be shown.

*P. aeruginosa* harbors a large variety of type II TA systems located on the chromosome, plasmids or prophages (Li et al, 2023). However, only four of these, including NatT, are part of the core genome (Andersen et al, 2017). Similar to other RES domain toxins (Freire et al, 2019), *natT* expression is induced by different stressors, including redox stress, antibiotics or in media mimicking host conditions (Teitzel et al, 2006; Zadeh et al, 2022; Song et al, 2023). Moreover, a *P. aeruginosa natT* mutant was recently shown

to be more susceptible to killing by different classes of bactericidal antibiotics and to promote survival in macrophages (Song et al, 2023). This is reminiscent of the proposed role for the RES domain toxin MbcT in the survival of *M. tuberculosis* in macrophages (Freire et al, 2019). A specific role in host persistence could also explain our findings that NatT modulates host killing in a simple insect larvae infection model. Thus, although the physiological cues regulating NatT expression and activity remain to be defined, this TA system seems to play an important role in *P. aeruginosa* homeostasis in the host. In line with this, we have identified *natT* alleles in patient isolates displaying mild toxin activation and increased antibiotic tolerance. It is possible that NatT function is under selection during infections to increase basal level persistence provided by this TA system.

NatT function is tightly controlled by NatR, which is both an anti-toxin and an auxiliary factor required for toxin stability. Genetic, biochemical and biophysical analyses suggested that the NatR–NatT complex is able to switch between an inert hexameric state (4R-2T) and an active 2R-T complex. Because only the hexameric complex is able to bind DNA, we propose that NatT activity is coupled to transcriptional autoregulation through this oligomeric switch. The NatT Flap provides a large interaction surface for the anti-toxin, effectively locking NatR' and its C-terminal helix in the NatT active site cavity. NatT activation likely includes the disintegration of the NatT dimer, which in turn would liberate the Flap and allow NatR' to move away from the active site cavity without disengaging from its toxin partner. Residue E29 interacts with the C-terminal helix α7 of NatR' and, through its immediate neighbor Q30, also participates in NatT dimerization. Thus, the activating mutation E29D may not only disturb the interaction between the Flap and helix α7, but may also disrupt NatT dimerization. If and how the phosphate molecule positioned at the Flap-NatR' interface influences NatR–NatT interaction, remains to be shown. Residues coordinating this phosphate are highly conserved, arguing that it has likely adopted an important structural or mechanistic role. Although this pocket is not accessible in the fully closed hexameric state, it is possible that rearrangement of Flap and NatR' in the active conformation provides access for phosphorylated metabolites that regulate NatT activity. A possible candidate is ADPR-1P, one of the NAD breakdown products generated by NatT. Positive feedback mediated by this unique metabolite could explain why a catalytic NatT mutant (R82A) was unable stably derepress *natR–natT* transcription. In agreement with NatT and NatR forming a stable interaction through the Flap, NatT is rapidly degraded in *P. aeruginosa* strains lacking NatR and, in contrast to RES domain toxins lacking a Flap (Skjerning et al, 2019; Freire et al, 2019; Milunovic et al, 2014), showed no toxicity by itself or when expressed in *E. coli*. Genome analyses have indicated that the Flap region was adopted by RES domain toxins several times independently, arguing that it marks an important evolutionary step towards more sophisticated toxin control by their cognate antitoxins.

Although this work has revealed exciting new mechanistic insights, the signals or physiological cues responsible for NatT toxin activation and its downstream consequences remain to be defined. Several important aspects need attention. (1) NatT toxin activity was limited to a small subpopulation of cells even in strains constitutively expressing *natR* and *natT*. Which auxiliary factors or mechanisms are

responsible for this stochastic control? (2) Derepression of *natR–natT* transcription requires NatT catalytic activity. How is NAD degradation coupled to *nat* gene expression? (3) Supplementing media with the NAD precursor nicotinamide effectively blocked NatT activity and expression. How do metabolic cues feedback on the NatR–NatT system and could this help explain its stochastic nature? While this effect could be indirect, it is also possible that NatT is directly inhibited by its product, similar to the mammalian TIR domain NADase SARM1, a central switch in axon degeneration (Bratkowski et al, 2020). (4) Similar to MbcT from *M. tuberculosis* (Freire et al, 2019), NatT is an NAD phosphorylase converting NAD into Nam and ADPR-1P. ADPR-1P is known to be produced during tRNA splicing (Banerjee et al, 2019) in eukaryotes, but has not been implicated in any other cellular processes. Does ADPR-1P have a specific role in mediating NatT downstream effects distinct from ADP-ribose (ADPR), the breakdown product of physiological NAD hydrolases? And does the salvage pathway repressor NrtR distinguish between ADPR-1P and its cognate ligand ADPR (Okon et al, 2017)? TIR domain NADases involved in cell death signaling and abortive infection produce cyclic derivatives of ADP-ribose as potent signaling molecules (Bayless et al, 2023; Ofir et al, 2021). It will be interesting to test if ADPR-1P has similar properties or if RES domain phosphorylases operate on the simple logic of depleting $NAD^+$ pools.

# Methods

## Bacterial strains and culture conditions

Strains used in this study are listed in Appendix Table S1. Unless otherwise stated, *P. aeruginosa* and all *E. coli* strains were grown at 37 °C in Luria Bertani (LB) medium under shaking at 170 rpm or on plates containing 1.3% agar where appropriate. Media *P. aeruginosa* were supplemented with 30 μg/ml gentamycin (*E. coli* 20 μg/ml) or 100 μg/ml tetracycline (*E. coli* 12.5 μg/ml). Stocks of 1 M isopropyl-β-D-thiogalactopyranoside (IPTG), 1 M nicotinamide (NAM), 10 mg/ml tobramycin, 1 mg/ml Ciprofloxacin were prepared in water. Stocks of 1 M cumate and 30 mg/ml Chloramphenicol were prepared in 100% ethanol.

## Plasmids and oligonucleotides

Plasmid and oligos used in this study are listed in Appendix Tables S2 and S3.

## Molecular biology procedures

Cloning was carried out as previously described (Manner et al, 2023). DNA fragments were amplified by PCR using Phusion polymerase. Vectors were cut by restriction and dephosphorylated using calf intestinal alkaline phosphatase. Vectors and inserts were gel purified and ligated at 16 °C overnight or room temperature for 10 min using T4 DNA Ligase. After ligase deactivation at 65 °C for 10 min, chemically competent *E. coli* DH5a cells were transformed by heat shock at 42 °C for 45 s, followed by phenotypic expression and plating on LB agar containing the appropriate antibiotic. Constructs were validated by sequencing. Details for each construct are provided in Appendix Table S1. Primers used in this study are listed in Appendix Table S3.

## Chromosomal deletion by homologous recombination

All chromosomal deletions and allelic replacements were engineered via two-step allelic exchange using pEX18-based vectors (Choi and Schweizer, 2005). For the pEX18Tc-based (Tet$^R$) deletion constructs flanking regions (c.a. 700 bp) of the target gene were cloned. For pEX18Tc-based constructs to generate chromosomal *gfp* reporter strains, the eGFP gene was cloned between regions flanking the target gene. *Pseudomonas aeruginosa* was transformed by electroporation, resistant colonies selected by picking onto selective plates and then transferred to plates containing 8% sucrose to select for double cross-over. Mutants were validated by colony PCR (deletions, insertions) or by sequencing (allelic exchange). The plasmids used are listed in Appendix Table S2.

## Site-directed mutagenesis

Individual alleles *of natT* were constructed using the Quick-Change mutagenesis protocol (Stratagene) with Phusion DNA polymerase (NEB) and the respective oligonucleotides according to the manufacturer's instructions (GE Healthcare). After amplifying the vectors using the mutagenesis primers listed in Appendix Table S3, target vectors were digested for 1 h at 37 °C using *Dpn*I restriction endonuclease. Point mutations were verified by DNA sequencing.

## Ectopic expression of *natT* alleles in *P. aeruginosa*

To express different alleles of *natT* or *natR* in *P. aeruginosa*, strains carrying the empty plasmid pME6032 or pME6032 harboring *natT* or *natR* copies were grown ON in LB supplemented with Tet 100 μg/ml and then diluted to an OD 0.05 with fresh LB supplemented with different concentrations of IPTG (0–250 μM) and grown in microtiter plates in an Epoch-2 reader (Agilent Technologies). 20 mM NAM was supplemented where indicated.

## Antibiotic killing assays

To quantify *P. aeruginosa* survival during antibiotic treatment, cultures were inoculated from a single colony in LB or in LB supplemented with NAM, grown to different ODs as indicated, and then diluted back with fresh LB medium containing antibiotics to a standard OD of 0.12. Unless otherwise stated, tobramycin and ciprofloxacin were used at 16 μg/ml and 2.5 μg/ml, respectively. Culture aliquots were collected at regular time intervals, diluted, and plated on LB plates to determine survival frequencies. *P. aeruginosa* strains carrying derivatives of plasmid pME6032 carrying different *natR* or *natT* alleles were grown in LB supplemented with 100 μg/ml tetracycline, diluted to OD 0.1 in LB Tet 100 μg/ml with or without 250 μM IPTG and grown for four hours to allow the expression of the protein. Bacteria were then diluted 1:2 in LB supplemented with 16 μg/ml tobramycin. Culture aliquots were collected at regular time intervals, diluted, and plated on LB plates to determine survival frequencies.

## Determination of minimal inhibitory concentrations (MIC)

MIC values were determined by adapting existing protocols (Wiegand et al, 2008; Liebens et al, 2017). Briefly, an overnight culture was diluted in LB to reach a density of $10^6$ CFU/ml, and

grown in the presence of increasing concentrations of antibiotics for 16–20 h at 37 °C with shaking. After incubation, the $OD_{600}$ was determined and MIC values were determined as the lowest antibiotic concentration where no growth was observed.

## Competition experiments

Two different strains constitutively expressing eGFP and mCherry, respectively, were mixed 1:1 in 5 ml LB medium and cultures were grown to stationary phase ON. Cultures were then diluted back 1:100 into fresh LB growth cycles were repeated for 3 days. Culture aliquots were removed daily, diluted in PBS and used to determine their composition by flow cytometry.

## Flow cytometry

To determine subpopulations with different lag periods during outgrowth, strains constitutively expressing TIMER[bac] (Claudi et al, 2014) from the chromosome were inoculated in LB and grown into stationary phase. Bacteria were diluted with fresh LB to an OD of 0.12, incubated for 3 h before the fraction of green and red cells was determined by flow cytometry. To determine *natT* expression or *pncB1* expression in individual cells, cultures of strains harboring a chromosomal *natT::gfp* or *pncB1::gfp* fusion were inoculated in LB medium. At the indicated time points culture aliquots were removed, diluted in PBS, and analyzed by flow cytometry. Flow cytometry measurements were performed on a flow cytometry Fortessa and fluorescence determined with the following channel, Ex488_LP495_BP514/30-H for eGFP and Ex561_LP600_BP610/20-H for mCherry.

## FACS sorting

Cells expressing TIMER[bac] (Claudi et al, 2014) were inoculated in LB and grown into stationary phase, before being diluted back to an OD of 0.12 with fresh LB or LB with 20 mM NAM, and grown for three hours. Cell cultures were sorted according to their GFP/mCherry ratio using an Aria (BD Biosciences) with scatter and fluorescence channels (green, Ex488_LP495_BP514/30-H; mCherry, Ex561_LP600_BP610/20-H). Triplicates of sorted cells ($4 \times 10^5$) were inoculated in 1 ml LB containing tobramycin (10 mg/ml) and incubated for different time intervals before aliquots were removed and plated on LB plates to determine survival rates. Aliquots of sorted populations were directly plated on LB plates to determine the number of viable bacteria after sorting.

## Fluorescence microscopy

Overnight cultures were diluted 1:100 in fresh media and transferred to 1% agarose pads containing LB medium and IPTG where indicated. Time lapses were recorded on a DeltaVision microscope (Applied Precision) equipped with a ×100 oil immersion objective and an environmental chamber maintained at 35 °C. Images were recorded using a CoolSnap HQ2 camera and processed using Softworx software (Applied Precision).

## Immunoblot analysis

Strains expressing FLAG-tagged versions of NatT were inoculated in LB medium or in LB supplemented with 20 mM NAM and harvested in different growth phases. Aliquots were sampled at different times points, pelleted by centrifugation and snap-frozen in liquid nitrogen. Samples were dissolved in SDS sample buffer containing 1% beta-mercaptoethanol and 2% SDS, boiled for 5 min and separated by SDS-PAGE (15%). Proteins were semi-dry transferred to a nitrocellulose membrane (Amersham Protan 0.2 μm), and membranes were blocked overnight in PMT (PBS, 0.1% v/v Tween 20, 5% w/v dry milk) and stained with polyclonal anti-FLAG M2 antibodies (1:10,000) (Sigma) and with polyclonal swine anti-mouse-HRP antibodies (1:1000, Dako, Denmark). Protein bands were stained with ECL reagents, and chemiluminescence was detected with a LAS4000 imager using ImageQuant LAS4000 version 1.3.

## Metabolome analysis

Cells were pelleted and snap-frozen in liquid nitrogen. Metabolites were extracted twice with hot (>70 °C) 60% ethanol. Extracts were analyzed by flow injection—time of flight mass spectrometry on an Agilent 6550 QTOF instrument operated in the negative mode, as described previously (Fuhrer et al, 2011).

## Quantification of NAD and NADP

The concentrations of $NAD^+$/NADH and $NADP^+$/NADPH were determined using Assay Kits from Fluorometric following the manufacturer's instructions. Overnight cultures of *P. aeruginosa* were diluted 1:100 in LB, grown to different optical densities before aliquots were removed, washed twice in PBS and resuspended in the lysis buffer provided by the manufacturer.

## Protein expression and purification

Overnight cultures of *E. coli* BL21 (DE3) (Thermo Scientific) carrying a pRSF-Duet-1 plasmid with *natR* and a His-tagged copy of *natT* were diluted 100× in 4 L of fresh LB medium supplemented with kanamycin at 100 μg/ml. Expression was induced by the addition of 500 μM IPTG at an OD of 0.8 for 16 h at 18 °C. Cells were harvested by centrifugation at $7000 \times g$ for 15 min, and pellets frozen at −80 °C. Before lysis, the cell pellet was resuspended in 50 ml of a solution containing 50 mM Bicine pH 8.5, 300 mM KCl, 10 mM TCEP, 0.5 mg/ml lysozyme and protease inhibitor (cOmplete, Mini, EDTA-free, Roche). Lysis was performed by three passages through a cell disruptor, and the insoluble fraction was separated from soluble protein by ultracentrifugation for 30 min at $135,000 \times g$. The supernatant was loaded on a HisTrap column HP 5 ml (Cytiva) equilibrated with 50 mM Bicine pH 8.5, 300 mM KCl, 10 mM TCEP, and 20 mM Imidazole. Proteins were eluted by an imidazole gradient (20–500 mM) in 15 column volumes. Fractions were analyzed SDS-PAGE gel to assess protein quality and purity. Protein fractions were pooled, concentrated and submitted to a size exclusion chromatography S200 16/60 (Cytiva) at a flow of 0.5 ml/min in 50 mM Bicine pH 8.5, 50 mM KCl, 10 mM TCEP. Protein quality was again assessed by SDS-PAGE and fractions corresponding to the NatR–NatT complex were concentrated to 3–5 mg/ml, aliquoted and flash-frozen with liquid nitrogen and stored at −80 °C. The same protocol was applied for $NatT_{E29D}$ mutant protein complexes.

NatT samples for enzymatic assays were obtained directly from *P. aeruginosa* strains carrying plasmid pME6032 expressing FLAG-

tagged versions of *natT* or *natT*$_{E29D}$. *P. aeruginosa* cultures (500 ml) were induced with IPTG (500 μM) for 3 h at 37 °C at OD 0.8. Cells were harvested by centrifugation at 7000 × *g* for 15 min and were lysed in 10 ml PBS supplemented with protease inhibitor (cOmplete, Mini, EDTA-free, Roche). NatT was purified by immunoprecipitation using Anti-FLAG magnetic beads (Merk) and eluted with FLAG peptide following the manufacturer's instructions. Protein quality was assessed by SDS-PAGE and identical protocols were applied to NatT and NatT$_{E29D}$.

## Mass photometry analysis of NatRT complex and NatRT complex with dsDNA

Mass photometry was carried out on a Refeyn OneMP instrument (Refeyn Ltd.) in microscope coverslips of 24 × 50 mm in size and 1.5 mm thickness (Marienfeld, Cat. No. 0107222). The coverslips were washed with water, 50% isopropanol, and dried using compressed air according to the manufacturer's instructions. In order to form drops for the measurement, silicone gaskets were used on top of the coverslips (Merck, GBL103250). The instrument was calibrated with molecular weight markers (NativeMark™ Unstained Protein standard, Invitrogen, Cat. No. LC0725). Focus for the measurement was found using 18 μl of buffer without protein (50 mM Bicine pH 8.5, 50 mM KCl). After locking the focus for the measurement, 2 μl of the sample was added to the initial 18 μl of the buffer. Measurements were performed with NatR–NatT complexes at a final concentration of 10 nM. For DNA binding studies, dsDNA corresponding to the *natR* promoter region was added to the measurements with the protein complex at a final concentration of 1 μM. Movies were recorded for 60 s using Refeyn Acquire$^{MP}$ software (version 2.5.1) and analyzed with Refeyn Discover$^{MP}$ (version 2.5.0).

## NatR–NatT crystallization and data collection

Se-Met crystals of NatR–NatT complexes were produced by sitting drop vapor diffusion method in a crystallization solution composed of 0.25 M di-Ammonium tartrate and 17.8% PEG 3350, optimized from the condition n° 86 of the commercial screen NeXtal PEG suit HT (Molecular dimensions). NatR–NatT was crystallized in 0.2 M Sodium sulfate 0.1 M Bis-Tris propane 7.5 20% w/v PEG 3350 from the crystallization screen Pact Premier (Molecular dimensions) at a final protein concentration of 2.5 mg/ml. NatR–NatT$_{E29D}$ crystals were formed in 0.2 M sodium/potassium phosphate pH 7.5, 0.1 M HEPES 7.5, 22.5% v/v, PEG Smear Medium, 10% v/v Glycerol from condition G11 of BCS screen (Molecular dimensions) at a final concentration of 2.5 mg/ml. The crystallization plates were set by a Gryphon robot (Art Robbins Instruments) and stored at 20 °C. All crystals appeared between 4 and 7 days in drops prepared at 1:1 v/v protein:crystallization solution in a total volume of 0.4 μl.

## Data collection and structure determination

Data were collected at the Swiss Light Source (SLS), Villigen, Switzerland, at beamline X06DA - PXIII using DA + data acquisition software and were processed by XDS and CCP4i2 suite. Se-Met NatR–NatT crystals were diffracted at 3.2 Å, and an initial structure was obtained by single-wavelength anomalous dispersion (SAD) using Crank2 software. Native NatR–NatT crystals were diffracted at 2.3 and 2.4 Å, and the structure was solved by molecular replacement using the preliminary structure obtained from SAD. Model building was carried out on COOT, and refinement was performed using Phenix. Structure figures were prepared in ChimeraX. Statistics of the crystallographic data and protein refinement are shown in Appendix Table S4.

## NAD⁺ phosphorylase assay

Purified NatR–NatT$_{E29D}$ complex at a concentration of 40 nM was used in enzymatic assays with 5 mM of NAD⁺ in 25 mM Tris pH 8.0, 300 mM KCl in the presence or absence of 20 mM Potassium phosphate in a total volume of 500 μl. Reactions were started by adding the enzyme to a microcentrifuge tube, and the content was immediately transferred to an NMR tube. For end-point products analysis, the reactions were carried out for 16 h and 1D ¹H, 1D ³¹P, and 2D [³¹P,¹H]-HMBC NMR spectra were recorded. For the determination of the NatT kinetic parameters a series of 1D ¹H spectra were recorded sequentially (20 min per experiment), and the concentration of NAD⁺ was determined by integration of the NMR signals. The NMR experiments were recorded at 298 K on a Bruker Ascend 600 MHz spectrometer running Topspin 3.2 equipped with a cryogenically cooled triple-resonance probe. All data were processed with Bruker TOPSPIN-NMR software (version 3.2, Bruker).

## Analysis of *natT* and *natR* in clinical isolates

Sequences of *natT* and *natR* were obtained from 9594 *Pseudomonas aeruginosa* raw genomic sequences from the Sequence Read Archive database (SRA) using blastn_vdb tool and *natT* from PAO1 as input sequence. A total of 8286 *natT* and 8266 *natR* sequences were identified. Raw reads were translated and SNPs were spotted. Variant *natT* alleles were amplified by colony PCR from a collection of 300 clinical isolates of *P. aeruginosa*. The resulting PCR products were sequenced and cloned into pME6032.

## Mother machine experiments

We used a dual-input mother machine (DIMM) developed by (Kaiser et al, 2018). This tool is freely available at Metafluidics (Kong et al, 2016), an open-source repository for fluidic systems. The construction of a 5″ quartz mask with chrome layer was outsourced to the Compugraphics Jena GmbH and stored by the company Microresist (Berlin) from which we ordered masters of 1 μm cell trap height and 20-μm flow layer height. In total, 100 μl of overnight culture from a single colony were diluted in 10 ml of LB in a 50 ml Erlenmeyer flask and was incubated at 37 °C. When the OD reached 1.0, 10 ml of the culture were centrifuged at 5000 rpm for 5 min at 24 °C and resuspended in 20 μl of fresh LB. The rest of the overnight culture was centrifuged, and the supernatant was filtered twice through a 25 μm pore-size filter and used as spent media in the experiments. To analyze the single-cell activity of the *P. aeruginosa natT*$_{ED29}$ mutant, experimental setups were divided into different phases: (1) Exponential, (2) Gradual to Stationary, (3) Stationary, (4) Wake-up, (5) Antibiotic treatment, and (6) Recovery. The exponential phase consisted of exposing the cells to fresh LB medium for 3 h. During the gradual introduction to the stationary phase, cells were exposed to different ratios of fresh LB and spent LB (SLB) for 2.5 h until the complete exposure to SLB for 10 h to mimic stationary phase conditions. During the drug

treatment phase, cells were exposed to LB containing tobramycin (16 μg/ml). The experiments ended with a recovery phase with fresh LB medium for 24 h.

## Microscopy and image acquisition

Microscopy images were acquired with a Nikon Eclipse Ti2 Inverted Microscope equipped with a Hamamatsu ORCA-Flash4.0 V3 Digital CMOS camera (C13440-20CU) and an objective100× NA1.45: Nikon Plan Apo Lambda 100x Oil Ph3 DM (MRD31905). The best-focusing performance was achieved with the Zeiss low fluorescence Immersion oil for fluorescence microscopy 518 F/37 °C Free, ne = 1518. Fluorescence imaging was acquired with LED light at 470 nm (for GFP) and 575 nm (for RFP) using a SPECTRA-X LED Illumination System as light source. Emitted light was filtered using a GFP/mCherry ET dual-band Chroma filter (F58-019).

## Data processing and analysis for single-cell segmentation and tracking

Cell segmentation and tracking of single-cells was done using the "Bacteria in Mother Machine Analyzer" (BACMMAN) (Ollion et al, 2019) and its deep-learning algorithm DistNet (Ollion and Ollion, 2020). Segmentation and tracking were manually corrected from BACMMAN kymographs using a Wacom tablet. Training and validation datasets were created in BACMMAN and extracted as HDF5 files. Training of DistNet was based on a Colab script provided by BACMMAN (https://colab.research.google.com/gist/jeanollion/d96bc2a8adef58563427932319bdee1b/finetune_distnet.ipynb). To increase the accuracy of the persisters cells segmentation and tracking, data curation and lineage reconstruction was done manually.

## Ethics statement

Clinical isolates of *P. aeruginosa* used in this study were cultured from patient samples collected for routine microbiological testing at the University Hospital, Basel. Sub-culturing and analysis of bacteria were performed anonymously. No additional procedures were carried out on patients. Cultures were sampled following regular procedures with written informed consent, in agreement with the guidelines of the Ethikkommission beider Basel EKBB.

## Data availability

NatR–NatT and NatR–NatT$_{E29D}$ models were deposited at the Protein Data Bank (PDB) under the accession codes 8QNL (https://doi.org/10.2210/pdb8qnl/pdb) and 8QNQ (https://doi.org/10.2210/pdb8qnq/pdb), respectively. Data analysis, lineage reconstruction, and plotting of single-cell measurements were done using a custom-made Python script that was deposited on GitHub (https://github.com/hector-ahg/EMBO_Journal_toxin-antitoxin_pseudomonas).

The source data of this paper are collected in the following database record: biostudies:S-SCDT-10_1038-S44318-024-00248-5.

## Peer review information

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

## Acknowledgements

The authors would like to thank the Swiss Light Source for crystallographic data collection, and Fabienne Hamburger for cloning several constructs used in this study. We also would like to acknowledge the experimental support of Tim Sharpe from the Biophysics facility of the Biozentrum, Oliver Biehlmaier from the Imaging Core facility of the Biozentrum, Alex Schmidt from the Proteomics Core facility of the Biozentrum, Stella Stefanova from the Biozentrum FACS Core Facility and Thomas Müntener from the Swiss High-field NMR Facility. This study was supported by the Swiss National Science Foundation project grant 310030_189253 to UJ and by the Swiss National Science Foundation NCCR grant 51NF40_180541 to UJ.

## Author contributions

**Isabella Santi**: Conceptualization; Data curation; Formal analysis; Investigation; Methodology; Writing—original draft. **Raphael Dias Teixeira**: Conceptualization; Data curation; Formal analysis; Investigation; Methodology; Writing—original draft; Writing—review and editing. **Pablo Manfredi**: Data curation; Formal analysis; Investigation; Writing—original draft. **Hector Hernandez Gonzalez**: Data curation; Formal analysis; Investigation; Methodology; Writing—review and editing. **Daniel C Spiess**: Data curation; Formal analysis; Investigation. **Guillaume Mas**: Data curation; Formal analysis; Methodology; Collected the NMR data and performed the NMR analysis. **Alexander Klotz**: Data curation; Formal analysis; Methodology. **Andreas Kaczmarczyk**: Data curation; Formal analysis. **Nicola Zamboni**: Data curation; Formal analysis; Funding acquisition. **Sebastian Hiller**: Conceptualization; Supervision; Funding acquisition; Writing—original draft; Project administration; Writing—review and editing. **Urs Jenal**: Conceptualization; Data curation; Formal analysis; Supervision; Funding acquisition; Investigation; Methodology; Writing—original draft; Project administration; Writing—review and editing.

Source data underlying figure panels in this paper may have individual authorship assigned. Where available, figure panel/source data authorship is listed in the following database record: biostudies:S-SCDT-10_1038-S44318-024-00248-5.

## Disclosure and competing interests statement

The authors declare no competing interests.

# Expanded View Figures

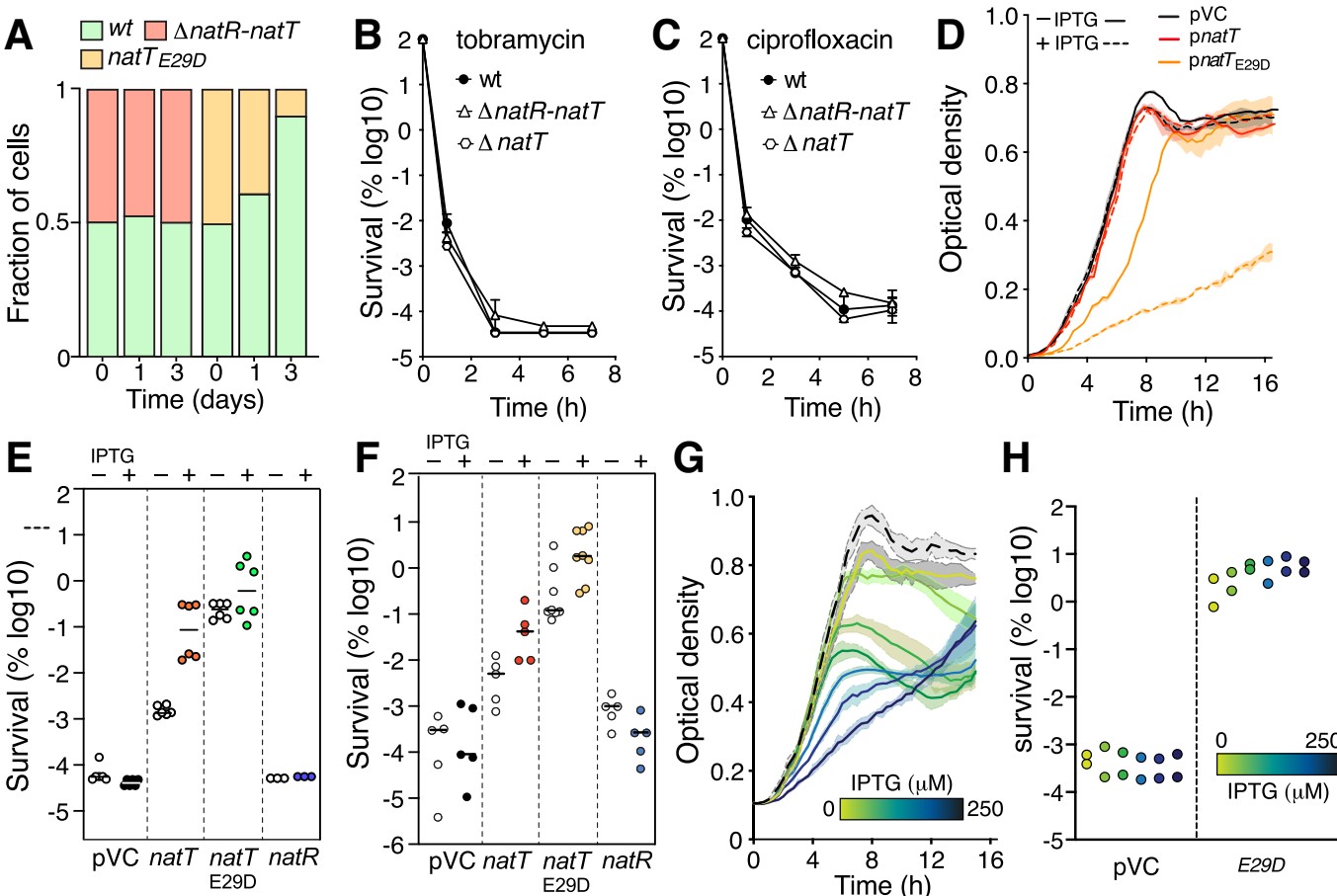

**Figure EV1. The NatT toxin confers drug tolerance to *P. aeruginosa*.**

(**A**) Reduced fitness of a *P. aeruginosa* natT_{E29D} mutant. *P. aeruginosa* strains expressing different fluorophores were mixed 1:1 and subjected to consecutive cycles of growth and re-dilution. Subpopulations were analyzed by flow cytometry at indicated time intervals. (**B, C**) Survival of ΔnatT and ΔnatR-ΔnatT mutants during treatment with tobramycin (8 μg/ml) (**B**) or ciprofloxacin (2.5 μg/ml) (**C**) (technical replicates, average ± SEM, *n* = 3). (**D**) Expression of *natT* impairs *P. aeruginosa* growth. Cultures of a ΔnatT mutant containing plasmids with IPTG-inducible *natT* alleles were grown in LB medium with or without IPTG (average ± SEM, *n* > 3) (pVC = control plasmid). (**E**) Expression of *natT* increases tolerance to tobramycin. Cultures of *P. aeruginosa* containing plasmids with IPTG-inducible *natT* or *natR* alleles were grown in LB medium without or with IPTG. Fractions of surviving cells were determined after three hours of treatment with tobramycin (16 μg/ml). Median values are indicted (*n* ≥ 3) (pVC = control plasmid). (**F**) Expression of *natT* increases tolerance to ciprofloxacin. Cultures of *P. aeruginosa* containing plasmids with IPTG-inducible *natT* or *natR* alleles were grown in LB medium without (empty boxes) or with 250 μM IPTG (filled boxes). Fractions of surviving cells were determined after three hours of treatment with ciprofloxacin (2.5 μg/ml) (*n* ≥ 5; lines mark medians) (pVC = control plasmid). (**G**) Expression of *natT_{E29D}* gradually limits *P. aeruginosa* growth. Cultures of a ΔnatT mutant containing a control plasmid (stippled line) or a plasmid expressing *natT_{E29D}* from an inducible promoter, were grown in LB with increasing concentrations of IPTG as indicated (average ± SEM, *n* > 3). (**H**) Expression of *natT_{E29D}* increases *P. aeruginosa* tolerance. *P. aeruginosa* cultures containing a plasmid with an IPTG-inducible *natT_{E29D}* were grown as in (**G**) and survival was determined after three hours of treatment with tobramycin (16 μg/ml) (average ± SEM).

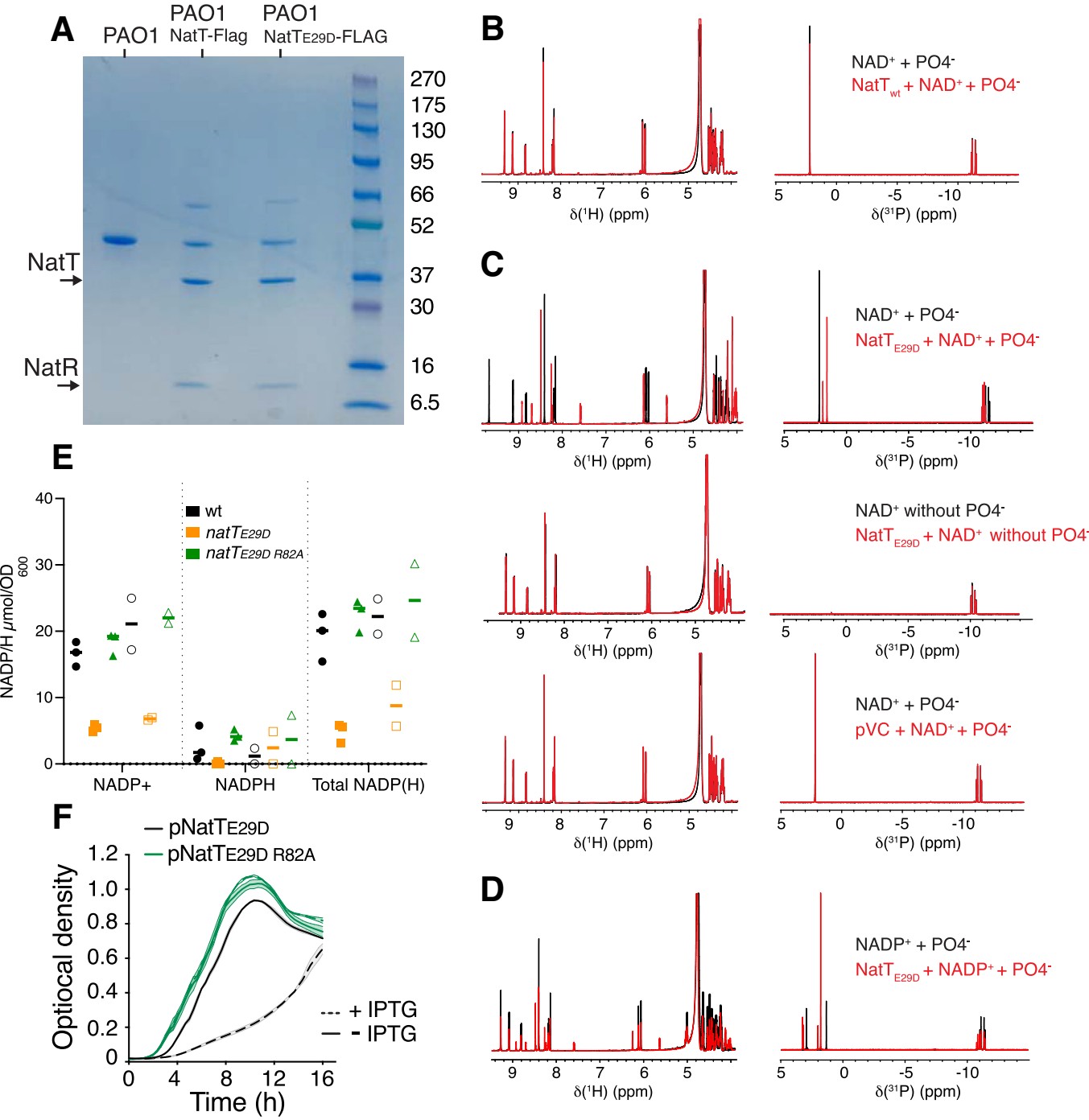

Figure EV2. NatT is a NAD-dependent phosphorylase.

(A) Purification of NatR–NatT complex from *P. aeruginosa*. NatR and NatT proteins were copurified from a *P. aeruginosa* Δ*natT* mutants harboring plasmids containing FLAG-tagged *natT* alleles by affinity chromatography using anti-FLAG beads. (B) Degradation of $NAD^+$ by $NatT_{E29D}$ is phosphate-dependent. Purified NatR–$NatT_{E29D}$ complex was incubated with $NAD^+$ with and without phosphate, and the reaction was analyzed by 1D $^1H$ and $^{31}P$ NMR spectra. (C) Purified NatR–NatT wild-type does not degrade $NAD^+$. (D) Purified NatR–$NatT_{E29D}$ degrades $NADP^+$ in a phosphate-dependent reaction. (E) NatT depletes cellular NADP pools of *P. aeruginosa*. $NADP^+$ and NADPH concentrations were determined in cultures of *P. aeruginosa* wild-type (wt) and in strains harboring plasmids expressing *natT_{E29D}* or *natT_{E29D R82A}* growing exponentially (filled bars) or from stationary phase (open bars) (average ± SEM, $n = 2$). (F) The active site residue R82 is required for NatT-mediated toxicity. *P. aeruginosa* with plasmids containing IPTG-inducible copies of *natT_{E29D}* (black line) or *natT_{E29D R82A}* (green line) were grown in LB with (dotted lines) or without IPTG (solid lines) (average ± SEM, $n = 3$).

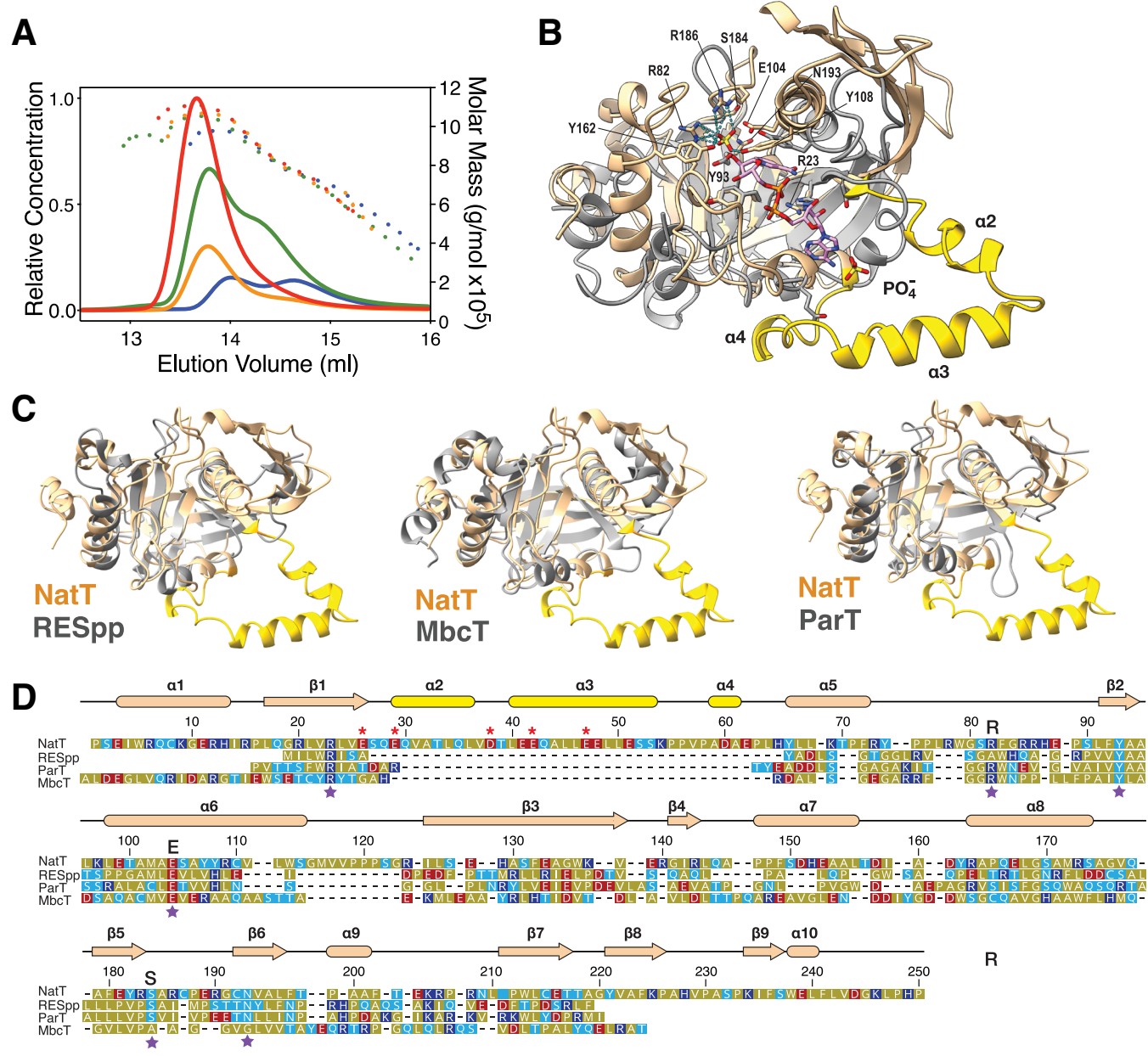

**Figure EV3. Interaction of NatR and NatT mediates toxin activation.**

(A) SEC-MALS analysis of NatR–NatT and NatR–NatT$_{E29D}$ complexes reveal different oligomeric structures. Experiments were performed with 7 µM (red) and 2 µM (orange) NatR–NatT and with 7 µM (green) and 2 µM (blue) NatR–NatT$_{E29D}$. (B) Structural homology between NatT and diphtheria toxin indicates analogous NAD$^+$ binding modes. NatT is shown in light brown and yellow (Flap), diphtheria toxin (1tox) is in gray and its active site NAD$^+$ is in pink. (C) Superposition of *P. aeruginosa* NatT (light brown) with Flap (yellow) and RES domain proteins RESpp (Skjerning et al, 2019), MbcT (Freire et al, 2019) and ParT (Piscotta et al, 2019) (gray). (D) Structure-guided sequence alignment of NatT and RES domain proteins shown in (B). Predicted active site residues are marked with purple stars. Charged residues of the Flap involved in interaction with NatR' are marked with red asterisks.

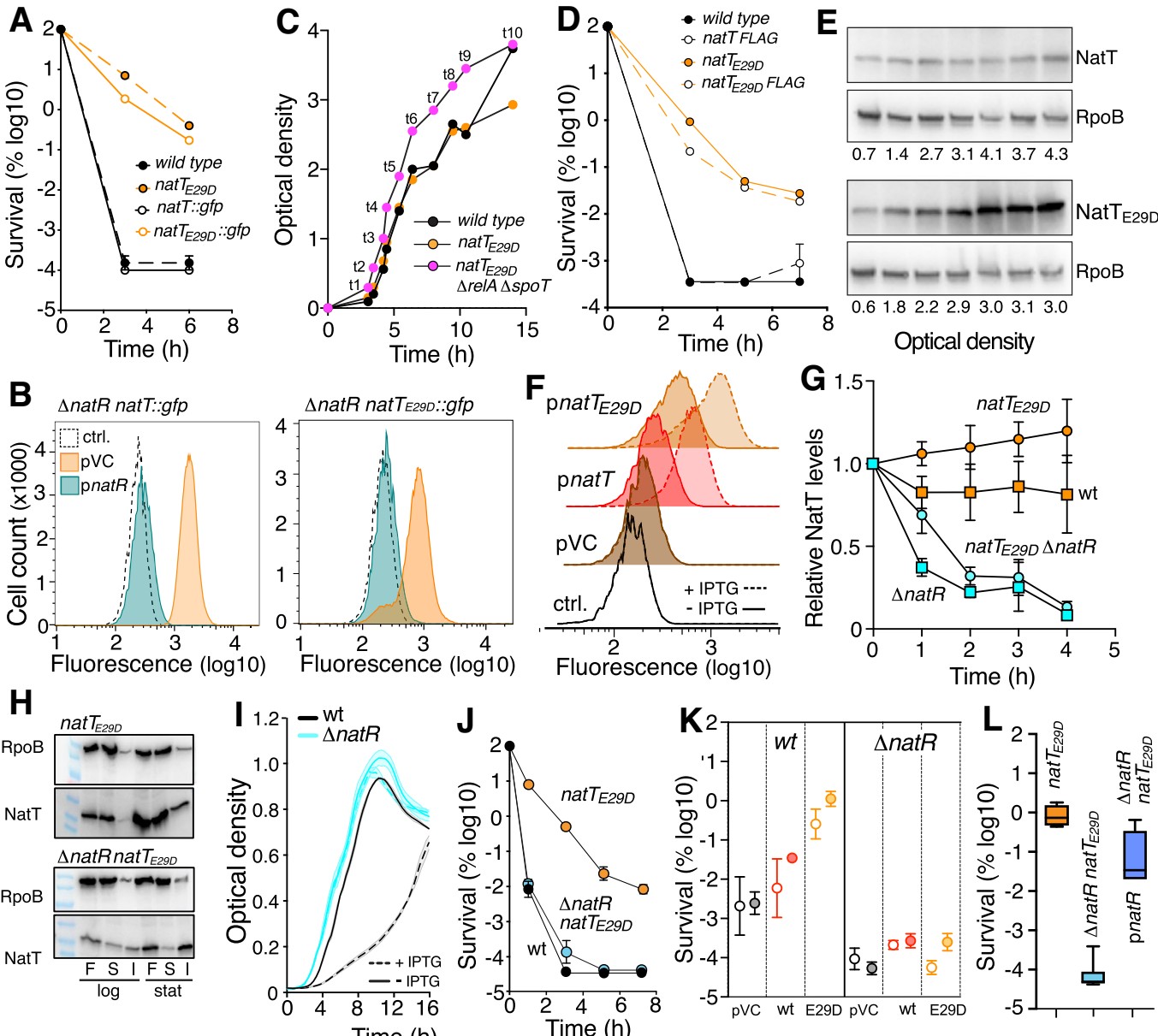

◄ **Figure EV4.  NatR is an anti- and a co-toxin of NatT.**

(A) Introduction of a chromosomal *natT::gfp* reporter does not affect *P. aeruginosa* drug tolerance. Survival of *P. aeruginosa* was scored during exposure to tobramycin (16 μg/ml) (technical replicates, average ± SEM, *n* > 3). (B) NatR is a repressor of *natT* transcription. Transcription of *natT* was determined by flow cytometry in Δ*natR* mutants with a *gfp* reporter downstream of *natT*. Strains contained a control plasmid (orange) or a plasmid expressing *natR* (green). Control strains lacking the *gfp* reporter are indicated by a dotted black line. (C) Growth of *P. aeruginosa* wild-type, *natT*$_{E29D}$ and *natT*$_{E29D}$ Δ*relA* Δ*spoT* in LB. Samples from different time points (t1-t10) were used for flow citometry analysis in Fig. 4E (*n* = 1). (D) Drug tolerance is not affected by an engineered chromosomal *natT-FLAG* allele (technical replicates, average ± SEM, *n* = 3). (E) NatT protein levels increase in *P. aeruginosa natT*$_{E29D}$ upon entry into stationary phase. Levels of NatT and NatT$_{E29D}$ were determined during growth (see: C) by immunoblot analysis using anti-FLAG antibodies and anti-RpoB antibodies as control. (F) Ectopic expression of *natT* leads to derepression of *natR–natT* transcription. Transcription of *natR–natT* was determined in *P. aeruginosa* harboring a chromosomal *natT::gfp* reporter and plasmids with IPTG-inducible *natT* alleles as indicated (pVC = control plasmid). (G) NatT is degraded in strains lacking NatR. Concentrations of NatT were determined by immunoblot analysis after treating *P. aeruginosa* cultures with chloramphenicol and plotted as relative values of the initial concentration (0 h) (technical replicates, average ± SEM, *n* ≥ 3). (H) NatT is insoluble in strains lacking NatR. NatT protein was quantified by immunoblot analysis of fractions harvested from different *P. aeruginosa* strains and in different growth phases as indicated. F: full lysate; S: soluble fraction; I: insoluble fraction. (I) NatT-mediated toxicity is abolished in a Δ*natR* mutant. Growth of *P. aeruginosa* wild-type and Δ*natR* mutant carrying a plasmid with an IPTG-inducible *natT*$_{E29D}$ allele was recorded in LB medium with or without IPTG as indicated (average ± SEM, *n* = 3). (J) NatR is required for NatT-mediated drug tolerance. Survival of *P. aeruginosa* wild-type and mutants indicated during exposure to tobramycin (average ± SEM, *n* > 3). (K) NatR is required for NatT-mediated drug tolerance. Survival of *P. aeruginosa* wild-type and Δ*natR* mutant carrying plasmids with IPTG-inducible copies of *natT* or *natT*$_{E29D}$ was determined after treatment with tobramycin (16 μg/ml). Cultures were grown with (filled circles) or without IPTG (empty circles) (average ± SEM, *n* = 3). (L) Ectopic expression of *natR* restores drug tolerance of a Δ*natR–natT*$_{E29D}$ mutant. Survival to tobramycin is shown for strains indicated. Plasmid p*natR* harbors an IPTG-inducible copy of *natR*. The box extends from the lower (25th percentile) to upper quartile (75th percentile) values, with a line at the median (50th percentile). Whiskers indicate the minimum and maximum values within 1.5 times the interquartile range (*n* = 3).

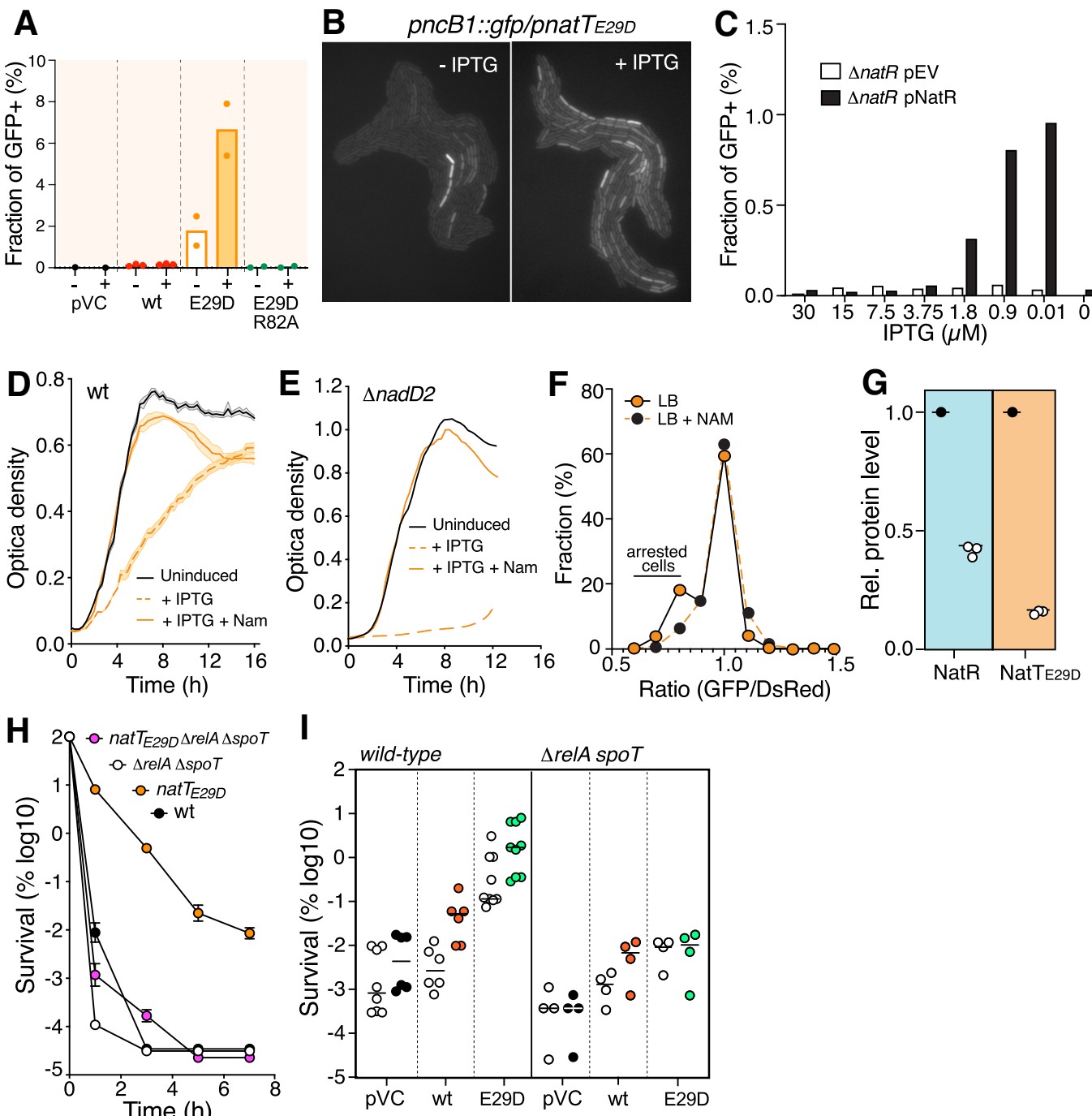

◀ **Figure EV5. The NAD salvage pathway neutralizes NatT toxin activity and abolishes drug tolerance.**

(A) Ectopic expression of *natT* mediates NAD$^+$ salvage pathway induction. Fractions of cells inducing salvage pathway genes were determined in *P. aeruginosa pncB1::gfp* reporter strains harboring plasmids expressing different *natT* alleles from an IPTG-inducible promoter. Cultures were grown with or without IPTG (average ± SEM, $n = 3$) (pVC control plasmid). (B) Ectopic expression *natT$_{E29D}$* induces salvage pathway genes. Representative microscopy images of a *pncB1::gfp* reporter strain expressing *natT$_{E29D}$* from a plasmid with or without IPTG. (C) Limiting *natR* expression induces salvage pathway genes. *P. aeruginosa ΔnrtR pncB1::gfp* carrying a plasmid with an IPTG-inducible *natR* was analyzed at different IPTG concentrations as indicated. Fractions of cells with derepressed salvage pathway were scored as a function of *natR* expression. (D, E) NAM neutralizes NatT$_{E29D}$-mediated growth defect in *P. aeruginosa* wild-type (D) or *ΔnadD2* salvage pathway mutant (E). Strains harboring a plasmid with an IPTG-inducible *natT$_{E29D}$* allele were grown with (orange) or without IPTG (black) and with (solid lines) or without NAM (20 mM) (dotted lines). (F) NAM overrides NatT-mediated growth arrest. Cultures of *P. aeruginosa* wild-type and *natT$_{E29D}$* mutant constitutively expressing TIMER were grown in LB with or without NAM for 3 h before analyzing populations by flow cytometry. Fractions of slow-growing or arrested cells were calculated using GFP/DsRed ratios of individual cells. (G) NAM limits NatR and NatT protein levels. A *P. aeruginosa natR$_{E29D}$* mutant was grown in LB with (open circles) or without NAM (closed circles), followed by the analysis of relative levels of NatR and NatT by mass spectrometry (technical replicates, average ± SEM, $n = 3$). (H, I) (p)ppGpp is required for NatT-mediated drug tolerance. (H) Cultures of *P. aeruginosa* strains indicated were treated with tobramycin and survival was determined over time (technical replicates, average ± SEM, $n = 3$). (I) Survival of *P. aeruginosa* wild-type and *ΔrelA ΔspoT* mutant carrying plasmids with IPTG-inducible *natT* (wt) or *natT$_{E29D}$* (E29D) alleles was determined after three hours of treatment with tobramycin. Cultures were grown with (filled circles) or without IPTG (open circles). Solid lines mark median values.

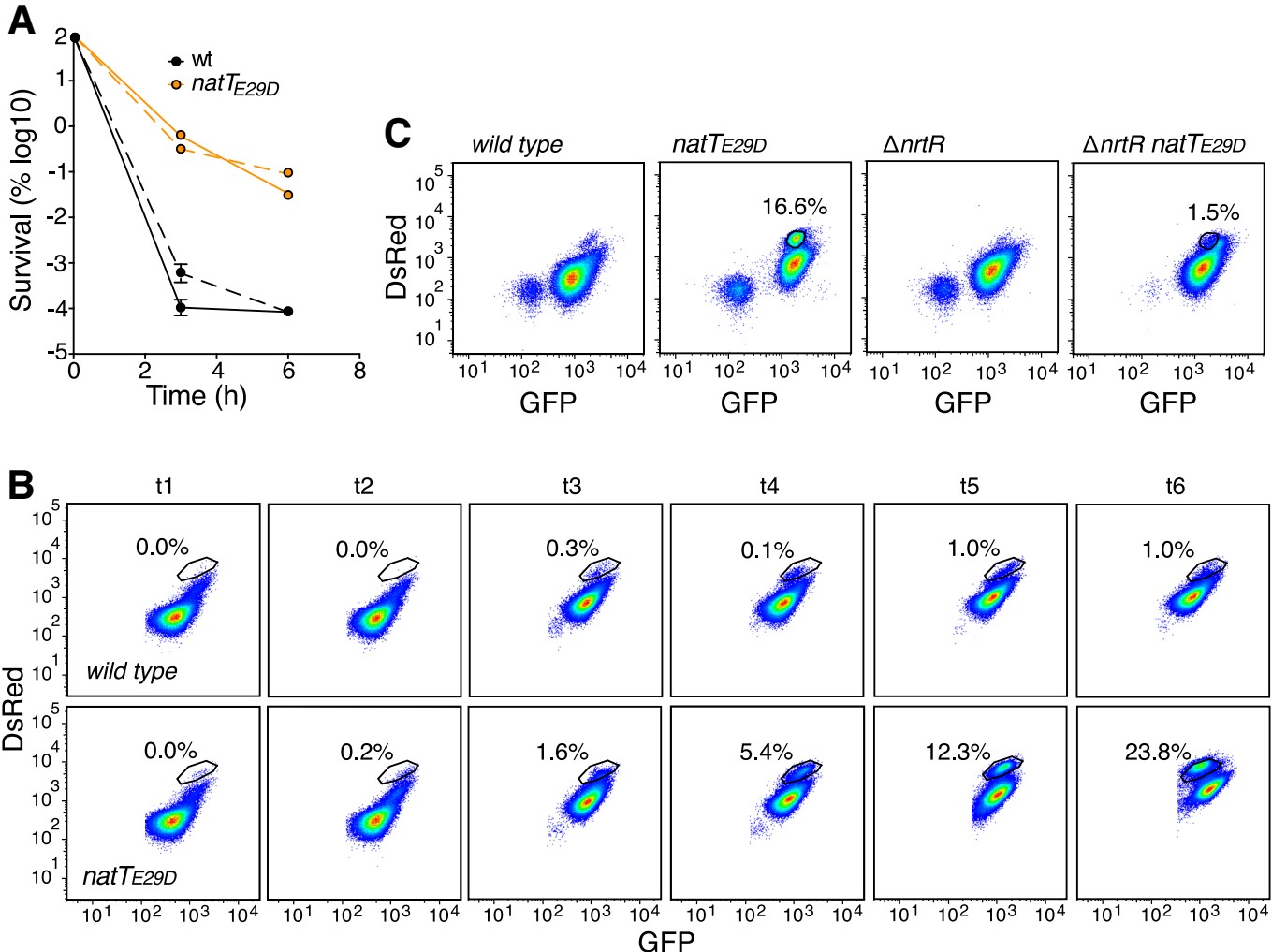

**Figure EV6.  NatT activation generates subpopulations with extended lag phase during outgrowth.**

(A) Expression of TIMER from the chromosomal *attb* locus does not affect *P. aeruginosa* drug tolerance. Cell survival during tobramycin treatment was determined for isogenic strains with (dotted line) or without (solid line) TIMER (technical replicates, average ± SEM, *n* = 3). (B) The *natT_{E29D}* allele induces growth arrest in a subpopulation of *P. aeruginosa* cells. Samples of *P. aeruginosa* wild-type and *natT_{E29D}* mutant cultures expressing TIMER were harvested at different time points (see: Fig. 6A), diluted into fresh medium for three hours, and analyzed by flow cytometry. Fractions of slow-growing cells (black gate) are indicated. (C) Deletion of *nrtR* abolishes the population of slow-growing *natT_{E29D}* mutant cells. Experiments were carried out as in (B) with cells being harvested at time point t5. Strains are indicated above the panels.

