## [Peer Review File · The EMBO Journal]

Toxin-mediated depletion of NAD and NADP drives persister formation in a human pathogen

Isabella Santi, Raphael Dias Teixeira, Pablo Manfredi, Hector Hernandez Gonzalez, Daniel Spiess, Guillaume Mas, Alexander Klotz, Andreas Kaczmarczyk, Nicola Zamboni, Sebastian Hiller, and Urs Jenal

Corresponding author: Urs Jenal (urs.jenal@unibas.ch)

Review Timeline:

Submission Date:	8th Mar 24
Editorial Decision:	11th Apr 24
Revision Received:	10th Jul 24
Editorial Decision:	8th Aug 24
Revision Received:	26th Aug 24
Accepted:	29th Aug 24

Editor: Ieva Gailite

Transaction Report:

Dear Urs,

Thank you for submitting your manuscript for consideration by the EMBO Journal. We have now received comments from two reviewers, which are included below for your information.

As you will see from the reports, both reviewers appreciate the topic and the quality of the work, while reviewer #1 is more positive on the broader novelty of the findings. Based on overall interest expressed in the reviewer reports, I invite you to revise the manuscript along the lines indicated in the referee comments. In particular, please focus on extending the manuscript along the lines indicated in the point 1 by reviewer #1 to strengthen the links between NatT activity, NAD⁺ degradation and resulting NAD salvage pathway activation, and bacterial growth arrest. I think it would be helpful to discuss the revision in more detail via email or phone/videoconferencing - please let me know which option you prefer.

We generally allow three months as standard revision time, which can be extended to six months in the case of major revisions. As a matter of policy, competing manuscripts published during this period will not negatively impact on our assessment of the conceptual advance presented by your study. However, please contact me as soon as possible upon publication of any related work to discuss the appropriate course of action. Should you foresee a problem in meeting this deadline, please let us know in advance to discuss an extension.

When preparing your letter of response to the referees' comments, please bear in mind that this will form part of the Review Process File and will therefore be available online to the community. For more details on our Transparent Editorial Process, please visit our website: <https://www.embopress.org/page/journal/14602075/authorguide#transparentprocess>. Please also see the attached instructions for further guidelines on preparation of the revised manuscript.

Please feel free to contact me if you have any further questions regarding the revision. Thank you for the opportunity to consider your work for publication. I look forward to discussing your revision.

With best regards,

leva

leva Gailite, PhD
Senior Scientific Editor
The EMBO Journal
Meyerohofstrasse 1
D-69117 Heidelberg
Tel: +4962218891309
i.gailite@embojournal.org

- a point-by-point response to the referees' comments, with a detailed description of the changes made (as a word file).
- a word file of the manuscript text.
- individual production quality figure files (one file per figure)
- a complete author checklist, which you can download from our author guidelines (<https://www.embopress.org/page/journal/14602075/authorguide>).

- Expanded View files (replacing Supplementary Information)

We realize that it is difficult to revise to a specific deadline. In the interest of protecting the conceptual advance provided by the work, we recommend a revision within 3 months (10th Jul 2024). Please discuss the revision progress ahead of this time with the editor if you require more time to complete the revisions.

Referee #1:

In this manuscript Santi, Dias Teixeira et al characterize the function, structure, and role of the hyperactivation of a toxin-antitoxin system in *P. aeruginosa* (Pa) encoded by PA1029-PA1030 (subsequently name natRT). In previous work the authors identified a hyper-persister mutation that mapped to natT (E29D), which they exploit here to better characterize the molecular mechanisms underlying the (dys)regulation and contribution of this TA system to antibiotic persistence in Pa. Using natT (E29D) they confirm that NatT is a NAD phosphorylase which depletes NAD⁺. They show that salvage NAD salvage pathways are induced by Pa in response to the activity of NatT and that this toxin promotes slow/arrested growth in a subset of the population, which in turn promotes survival to antibiotics.

This manuscript provides a substantial amount of data that certainly supports a role for natT (or at least natTE29D) in promoting a persister phenotype in Pa. This work is timely and important as the role of TA systems in bacterial growth arrest and the associated antibiotic tolerance has been heavily, and even disproportionately, disputed in recent years. This constitutes a thorough analysis of how the exaggeration of the activity of a TA because it is dysregulated (here by a point mutation in the toxin that blocks complete neutralization by the antitoxin and dysregulated the auto repression of the operon by the TA complex) can lead to the overrepresentation of one path to persistence in a pathogen. The work is therefore needed and interesting. A few experiments would strengthen the conclusion of the work and the presentation could be improved for the sake of clarity.

Major

1. Additional experiments should be provided to strengthen the causative nature of NatT activity, NAD⁺ depletion (and how this effects NAD salvage) on growth arrest. The identification of a slow growing population in the E29D mutant and its increased survival to Abs is very interesting. Does an increased induction of salvage eliminate this population for example in a double nrtR E29D mutant? Can the authors sort these sub-populations and test by qPCR the respective level of expression of salvage genes? Can they quantify the levels of NAD in these 2 populations? In addition, presumably NAD⁺ depletion leads to growth arrest by disrupting metabolism. Could the authors use a fluorescent reporter for redox or membrane potential and/or ATP content? This would go a long way in tying together all the parts the authors describe.

2. The authors must clarify in the relevant sections which toxin expression systems are being used (either plasmid or chromosomal and native or inducible promoter). In figure 1 for instance, IPTG inducible promoters have a high degree of leaky expression and for the E29D mutant in particular IPTG addition does little to increase survival relative its impact on growth (compared to the wildtype which shows no effect on growth with IPTG). As such it seems that there are instances where the effect on growth and Ab survival are uncoupled. Can the authors account for these differences and is using a single endpoint OD600 the best measure of this?

3. In several places description/explanations of experiments/results are lacking. This coupled with inconsistent figure panel ordering and labelling makes some sections very difficult to follow. For example, when discussing the link between NatT and NAD salvage it reads like what is being suggested is that it is a function of the toxin to induce salvage rather than because of the natural sensory capability of nrtR over the salvage operon. Despite the potential role of ADPR-1P leading to this being a hypothesis, the salvage pathway is upregulated because of the drop of NAD⁺ not because it is an explicit target of the toxin. This needs to be clarified earlier than in the discussion.

Minor

1. The effect that NAM has on various phenotypes is clear however there are some areas where the proposal that this feeds into salvage to protect is not very strong. For instance, in 5a *pcnA* deletion has no effect on growth. The authors should test the individual salvage mutants for Ab survival. Or show that intermediate survival of the *nrtR* mutant increases with Nam or a salvage mutant not responding to NAM. Could the authors supplement with other NAD intermediates?
2. Fig1h: It seems that the E29D mutant can get to stationary phase (based on OD600) then loses viability. It might be easier to demonstrate these growth phase effects by displaying CFU/mL relative to OD600 particularly in fig1.
3. Figure 3: do mutations of other flap residues besides E29D (further from active site) lead to hyperactive toxin too? This may allow the authors to decipher the effect of lack of complete neutralization vs lack of repression.
4. The organization of Figures 4 and 4s is very confusing.
5. Authors should better highlight the small population referred to in Figure 5b

Referee #2:

Santi and colleagues report an extensive structure-functional characterisation of *Pseudomonas aeruginosa* type II toxin-antitoxin pair NatR:NatT, earlier known as PA1030/PA1029 (Song et al, 2023). This is a rather understudied TA system. Song and colleagues previously used *E. coli* as a host and do a classical toxicity-neutralisation assay with wt and catalytically dead NatT. In their experiments NatT was toxic (but not very), and it was neutralised by co-expression of NatR. The weak toxicity is likely to be, as argued here by Santi and colleagues, due to rapid degradation of NatT due to poor folding.

The current study establishes that, as expected for RES toxins, NatT degrades NAD⁺. This was earlier shown for other RES-containing TAs, e.g. see (Freire et al, 2019; Skjerning et al, 2019). The authors solve X-ray structures of NatR:NatT and rationalise the gain-of-function E29D NatR mutation that results in NatT activation. This mutant was identified by the authors as associated with increased levels of antibiotic persistence, behaving analogously to classic *hipA7* allele (Korch et al, 2003). It is a solid study that opens up multiple follow-up avenues of research, with the most important being how is NatR:NatT activity induced? What is the trigger? The function of the wild-type NatR:NatT system under normal conditions is unclear, given that we do not know how the system is triggered. What does the genomic neighbourhood say? The authors mention that it is part of the core genome. Who are the neighbours?

This is a solid piece of work that uses microbiology, biochemistry and structural biology to characterise the NatR:NatT system extensively. I would recommend covering (Song et al., 2023) more extensively in the introduction - right now it is reference 30, and the toxicity-neutralisation assays from that paper are not mentioned at all. This would make the reader's life much easier. Similarly, more extensive comparison with other RES-containing TAs is warranted, e.g. (Freire et al., 2019; Skjerning et al., 2019). In general, when it comes to insights to molecular workings of RES-containing TAs, the study is adding useful new bits, but it is not ground-breaking - structures of this kind of TAs were reported before and the enzymatic activity is well-established. The interesting bit is the gain-of-function mutation of NatR that is analogous to famous *hipA7* allele in terms of increased persistence (Korch et al., 2003). I would expand on that parallel - currently *hipA7* is never mentioned. Overall, it is a solid piece of work but it unfortunately lacks when it comes to providing novel exciting biological insights.

References:

- Freire DM, Gutierrez C, Garza-Garcia A, Grabowska AD, Sala AJ, Ariyachaokun K, Panikova T, Beckham KSH, Colom A, Pogenberg V et al (2019) An NAD(+) Phosphorylase Toxin Triggers Mycobacterium tuberculosis Cell Death. *Mol Cell* 73: 1282-1291 e1288
- Korch SB, Henderson TA, Hill TM (2003) Characterization of the *hipA7* allele of *Escherichia coli* and evidence that high persistence is governed by (p)ppGpp synthesis. *Mol Microbiol* 50: 1199-1213
- Skjerning RB, Senissar M, Winther KS, Gerdes K, Brodersen DE (2019) The RES domain toxins of RES-Xre toxin-antitoxin modules induce cell stasis by degrading NAD⁺. *Mol Microbiol* 111: 221-236
- Song Y, Tang H, Bao R (2023) Comparative analysis of five type II TA systems identified in *Pseudomonas aeruginosa* reveals their contributions to persistence and intracellular survival. *Front Cell Infect Microbiol* 13: 1127786

Response to reviewers' comments

Referee #1:

In this manuscript Santi, Dias Teixeira et al characterize the function, structure, and role of the hyperactivation of a toxin-antitoxin system in *P. aeruginosa* (Pa) encoded by PA1029-PA1030 (subsequently name natRT). In previous work the authors identified a hyper-persister mutation that mapped to natT (E29D), which they exploit here to better characterize the molecular mechanisms underlying the (dys)regulation and contribution of this TA system to antibiotic persistence in Pa. Using natT (E29D) they confirm that NatT is a NAD phosphorylase which depletes NAD⁺. They show that salvage NAD salvage pathways are induced by Pa in response to the activity of NatT and that this toxin promotes slow/arrested growth in a subset of the population, which in turn promotes survival to antibiotics.

This manuscript provides a substantial amount of data that certainly supports a role for natT (or at least natTE29D) in promoting a persister phenotype in Pa. This work is timely and important as the role of TA systems in bacterial growth arrest and the associated antibiotic tolerance has been heavily, and even disproportionately, disputed in recent years. This constitutes a thorough analysis of how the exaggeration of the activity of a TA because it is dysregulated (here by a point mutation in the toxin that blocks complete neutralization by the antitoxin and dysregulated the auto repression of the operon by the TA complex) can lead to the overrepresentation of one path to persistence in a pathogen. The work is therefore needed and interesting. A few experiments would strengthen the conclusion of the work and the presentation could be improved for the sake of clarity.

We would like to thank this reviewer for carefully reading our work and for contributing constructive and valuable comments. We provide a point-by-point answer below.

Major

1. Additional experiments should be provided to strengthen the causative nature of NatT activity, NAD⁺ depletion (and how this effects NAD salvage) on growth arrest. The identification of a slow growing population in the E29D mutant and its increased survival to Abs is very interesting. Does an increased induction of salvage eliminate this population for example in a double DnrtR E29D mutant?

We show in Fig. 2 and Fig. S2, that NatT specifically mediates the degradation of NAD⁺ and NADP⁺ *in vitro* and that expression of NatT wild-type or the NatT_{E29D} mutant protein leads to the depletion of both co-factors *in vivo*. Moreover, we show in Fig. 5 and Fig. S5 that NatT expression leads to the derepression of the NAD⁺ salvage pathway and that NatT activity becomes highly toxic in mutants lacking enzymes of the NAD⁺ salvage pathway. We also show that combining the *natT*_{E29D} allele with a deletion of *nrtR*, the gene encoding the repressor of the NAD⁺ salvage pathway genes, largely abolishes (100x reduced) persister formation (Fig. 5g).

To further strengthen the causal relationship between NatT activity, NAD⁺ depletion and dormancy, we have added data to the revised manuscript (new Fig. S6c) demonstrating that the population of *natT*_{E29D} mutant cells with an extended lag is indeed strongly reduced in a $\Delta nrtR$ *natT*_{E29D} double mutant as compared to the *natT*_{E29D} single mutant. The observation that genetic derepression of the salvage pathway genes ($\Delta nrtR$) eliminates the population of cells with extended lags and, at the same time abolishes persister formation, strongly indicates that persister formation is linked to NAD⁺ depletion. Together,

these experiments make a convincing case for a causal relationship between NatT activity, NAD⁺ depletion and persister formation. We have added reference to these data on p.10:

“Deleting nrtR, the gene encoding the salvage pathway repressor, eliminated the fraction of natT_{E29D} mutant cells with extended lag phase almost entirely (Fig. S6c), indicating that metabolically dormant cells result from NatT-mediated NAD⁺ depletion. From this, we concluded that NatT activity generates arrested cells in stationary phase that experience extended lag periods when exposed to fresh nutrients.”

What we do not understand (see below) is how NatT activity and NAD⁺ shortages alter *P. aeruginosa* cell physiology in stationary phase to cause metabolic dormancy and extended lag periods, which eventually lead to antibiotic protection during outgrowth from stationary phase.

Can the authors sort these sub-populations and test by qPCR the respective level of expression of salvage genes? Can they quantify the levels of NAD in these 2 populations?

We would like to respond to these two points together, as they address the same aspect of persisters, namely how NatT activity and NAD⁺ degradation translate into persister formation. As emphasized above, we do not understand the mechanistic details of how NAD⁺ depletion influences cells in stationary phase in such a way that some of them remain metabolically dormant even after experiencing fresh nutrients during outgrowth from stationary phase. NatT-mediated persister formation strictly depends on cells experiencing nutrient depletion in stationary phase, but processes happening in stationary phase are difficult to access experimentally. It is possible that NatT-mediated NAD⁺ depletion directly leads to metabolic dormancy during stationary phase and by that to drug protection. Alternatively, NAD⁺ depletion may lead to persister formation indirectly, by inducing some form of metabolic stress (e.g., redox, energy status etc.) in nutrient depleted cells. To make this aspect clear to the reader, we have added the following statement to the discussion on p.15 (*“If persister formation is a direct consequence of NAD/P depletion or is caused indirectly by other forms of metabolic stress in nutrient deplete cells, remains to be shown”*).

The reviewer suggests different experiments that could help scrutinize these possibilities. We highly appreciate these ideas. To discuss the feasibility and significance of such experiments, it is important to stress again that experiments using the TIMER reporter do not specify ‘slow growing bacteria’ per se, but rather mark a subpopulation of cells with extended lag periods (stasis) during exit from stationary phase (Fig. 6 and Fig. S6). Because these cells do not show active growth, determining salvage pathway derepression or measuring NAD⁺ concentrations is technically challenging.

Salvage pathway genes: Experiments measuring mRNA levels of salvage pathway genes do have a major caveat in that bacterial mRNA is notoriously instable and is known to rapidly decay during FACS sorting. Attempts to execute robust RNASeq experiments with FACS sorted subpopulations have consistently failed in our hands. Alternatively, one could combine the TIMER growth reporter with the *pncB1* salvage pathway reporter to read out salvage pathway derepression during prolonged stasis. Because the TIMER reporter broadly spans fluorescence channels between green and red, adding a fully compatible fluorescent reporter for the salvage pathway genes is difficult. Finally, induction of salvage pathway genes is a good measure for stochastic NatT-mediated NAD⁺ depletion in growing cells but not in stationary phase or in metabolically dormant cells. About 3% of all cells show transient salvage pathway induction during exponential growth, but this number steadily declines when cells enter stationary phase, most likely because transcription of most genes rapidly stalls when nutrients become limited.

NAD⁺ concentration: As outlined above, using transcriptional reporters in non-growing cells as a proxy for NAD⁺ depletion is not possible. To circumvent these problems, we are in the process of adapting an allosteric reporter for NAD⁺ called SoNar (PMID: 25955212) to link cellular NAD⁺ to growth and persistence of individual *P. aeruginosa* cells. Allosteric reporters have several advantages, including the ability to measure concentration changes in real time and the fact that they do not rely on active transcription. Unfortunately, although investing considerable time and efforts in the past few months, we have not yet succeeded to tune this system to our needs, most likely because the dynamic range of SoNar is limited. Using *E. coli* and *P. aeruginosa* strains with an IPTG-inducible copy of well-characterized NADases like the necrotizing toxin (CpnT) from *Mycobacterium tuberculosis*, we have established assay strains, in which we can manipulate NAD⁺ levels at will. While we could demonstrate that CpnT toxin-induction leads to rapid growth arrest, the SoNar signal was not strong enough to robustly distinguish NAD⁺ levels under replete and deplete conditions. We also constructed SoNar-mScarlet-I fusions and found that while the mScarlet-I signal was strong and readily detectable, the 405 nm excitation/515 nm emission signal of SoNar was below detection limit. Although we are convinced that it is possible to adapt SoNar to determine NAD⁺ levels in non-growing cells by gradually increasing its signal strength and dynamic range, this will require more time and is thus beyond the possibilities of the revisions of the current manuscript.

In addition, presumably NAD⁺ depletion leads to growth arrest by disrupting metabolism. Could the authors use a fluorescent reporter for redox or membrane potential and/or ATP content? This would go a long way in tying together all the parts the authors describe.

As pointed out above, we do not understand how NAD⁺ shortages translate to metabolically inert persisters during stationary phase. Addressing this question will require more extensive experimental insight into the metabolic consequences of NatT-mediated NAD⁺ depletion. Although this is one of the key questions related to NAD⁺ degrading toxins and to TA systems in general, we feel that this aspect will need to be addressed in more targeted work in the future. It is important to emphasize that, despite decade-long investigations, a generally accepted metabolic or physiological mechanism causing extended lag periods and persister formation is still lacking in the field. Although shortages of key metabolites like ATP or different forms of stress have been proposed to be responsible for persister formation, convincing causal experimental evidence is outstanding.

While we feel that understanding the exact physiological details of how NAD⁺ depletion leads to persister formation is beyond the scope of this manuscript, we have added experiments to the revised version of the manuscript to strengthen the link between NatT activity, stationary phase physiology and persister formation at the single cell level. We have adopted a ‘dual input mother machine’ microfluidic setup that allows us to investigate growth and antibiotic killing of thousands of individual cells in small microchannels under changing media conditions. To mimic the different growth phases, cells were switched from continuous growth in rich media (LB) to spent media harvested from the supernatant of a stationary phase cultures of *P. aeruginosa* (SLB). By gradually replacing LB with SLB, bacteria experience a natural physiological transition into stationary phase. Antibiotic survival is then tested by switching starved *P. aeruginosa* cells back to fresh LB containing high antibiotic concentrations for 3 hours, after which surviving bacteria are monitored by drug washout and fresh media supply without antibiotics. Importantly, these experiments allowed us to monitor individual persisters and susceptible cells and link their antibiotic survival directly to their growth behavior before drug treatment. We have analyzed a total of over 120’000 individual cells, 241 of which being persisters.

These experiments are shown in the new Figs. 6d,e and f. Together, they demonstrate that 1) Similar to experiments with liquid cultures, persister formation is dependent on cells expressing an active variant of

NatT (E29D); 2) persister frequencies observed in the mother machine are similar to those observed in liquid cultures (about 1%); 3) persister formation strictly depends on bacteria experiencing stationary phase conditions. Cells that were not exposed to SLB failed to develop persisters; 4) persisters are characterized by a state of metabolic dormancy during outgrowth from stationary phase (red TIMER fluorescence) (Fig. 6d,f); and 5) the persister state is transient and is gradually lost within a few hours of outgrowth in fresh media (Fig. 6e). As seen in Fig. 6d, providing stationary cells with fresh medium triggers immediate growth resumption (switch of TIMER fluorescence from red to green) in the majority of cells, while roughly 1% of the cells remain metabolically dormant (red) and by that survive antibiotic treatment.

These analyses lay the foundation for follow-up studies to investigate the NatT-mediated physiological changes and metabolic limitations of future persisters in stationary phase.

To accommodate the new results and to make them best accessible to the reader, we have added an additional results chapter on p.10, termed “**NatT generates persister subpopulations with extended lag phase**”. All data described in this chapter are now summarized in Fig. 6 and Fig. S6.

2. The authors must clarify in the relevant sections which toxin expression systems are being used (either plasmid or chromosomal and native or inducible promoter). In figure 1 for instance, IPTG inducible promoters have a high degree of leaky expression and for the E29D mutant in particular IPTG addition does little to increase survival relative its impact on growth (compared to the wildtype which shows no effect on growth with IPTG). As such it seems that there are instances where the effect on growth and Ab survival are uncoupled. Can the authors account for these differences and is using a single endpoint OD600 the best measure of this?

We would like to thank this reviewer for raising this point, as it made us aware of potential ambiguities regarding growth and tolerance of strains with chromosomal and plasmid-driven *natT* alleles.

In general, we use plasmid-driven *natT* expression only to demonstrate population-wide NatT activity and its effect on growth (Fig. 1b), drug tolerance (Fig. 1c) or metabolism (Fig. 2c,d). This is necessary, as it allows to bypass the stochastic behavior of *natT* alleles expressed from the chromosome, which limits their phenotype to a small subpopulation of cells making more sophisticated single cell analysis tools mandatory. While such plasmid-based experiments are instructive to some extent, they have limited relevance regarding the physiological situation of NatT toxin control and activity. Also, as the reviewer rightly points out, under such conditions growth and survival can indeed be uncoupled, in that for instance higher expression of a *natT* allele leads to gradually reduced growth but not to more drug tolerance. There are several explanations for this observation. First, growth and antibiotic survival are not measured under the same physiological conditions. Please keep in mind that, because stationary phase is critical for persister formation, antibiotic survival is always scored during outgrowth from stationary phase. Second, very high levels of *natT* expression are toxic for *P. aeruginosa*, making it difficult to distinguish between stasis and cell lysis and making a direct correlation to persisters impossible.

To try to avoid confusion for the reader, we have moved Fig. 1b and 1c to the supplemental material (new Figs. S1d,e). Also, we have added information to the legends of figures 1 and S1 that makes it clear to the reader which expression systems were used for individual experiments.

3. In several places description/explanations of experiments/results are lacking. This coupled with inconsistent figure panel ordering and labelling makes some sections very difficult to follow. For example, when discussing the link between NatT and NAD salvage it reads like what is being suggested is that it is a function of the toxin to induce salvage rather than because of the natural sensory capability of nrtR over the salvage operon. Despite the potential role of ADPR-1P leading to this being a hypothesis, the salvage pathway is upregulated because of the drop of NAD⁺ not because it is an explicit target of the toxin. This needs to be clarified earlier than in the discussion.

We thank the reviewer for the suggestion and we hope the final version of the manuscript will have the panel order corrected. We have included two statements about salvage pathway regulation in the results (p.9).

“Transcription of the salvage pathway genes is controlled by NrtR, a repressor that binds ADP-ribose, one of the breakdown products of NAD⁺, leading to salvage pathway derepression³¹”.

Minor

1. The effect that NAM has on various phenotypes is clear however there are some areas where the proposal that this feeds into salvage to protect is not very strong. For instance, in 5a *pcnA* deletion has no effect on growth. The authors should test the individual salvage mutants for Ab survival. Or show that intermediate survival of the *nrtR* mutant increases with Nam or a salvage mutant not responding to NAM. Could the authors supplement with other NAD intermediates?

Please note that the experiments in Fig. 5a were carried out with plasmid-mediated *natT* expression. Although this experiment helped understanding the key role of the salvage pathway in countering NatT activity, it is not a physiological situation for survival (see main point 2, above). Under these conditions, deleting *pcnA* still had an effect on growth, although not as severe as deletions of other NAD⁺ biosynthesis genes like *pncB*, *nadD* or *nadE*. Why the expression of *natT_{E29D}* causes different degrees of growth inhibition in different mutant backgrounds, is unclear. Because paralogs exist for several genes involved in NAD⁺ synthesis, it is possible that PncA activity can be complemented by an unknown functional homolog. Alternatively, the observation that disruption of the terminal steps of NAD⁺ biosynthesis showed the strongest growth defect in cells with an active NatT argues that the *de novo* and salvage pathway branches of NAD⁺ synthesis may partially complement each other, or that specific precursors for individual catalytic reactions can also be provided by parallel metabolic pathways.

The role of nicotinamide (NAM) in blocking persister formation (Figs. 5d) remains unclear. Please note that adding NAM to the growth medium not only abolishes persister formation, but also leads to the complete repression of *natT* transcription (Fig. 5f). The observation that NAM neutralizes the physiological consequences of NatT even in the absence of a functional salvage pathway (Fig. S5e), indicated that it abolished persister formation by interfering with toxin expression and/or activity rather than by simply replenishing pools of nicotinamide dinucleotides. This exciting preliminary finding indicates that NatT activity may be regulated allosterically by NAM or other intermediates of NAD⁺ metabolism. Clearly, this will need to be scrutinized by additional experiments that go beyond the scope of this manuscript.

2. Fig1h: It seems that the E29D mutant can get to stationary phase (based on OD600) then loses viability. It might be easier to demonstrate these growth phase effects by displaying CFU/mL relative to OD600 particularly in fig1.

We believe that the data requested by the reviewer are already shown in Fig. 1f.

3. Figure 3: do mutations of other flap residues besides E29D (further from active site) lead to hyperactive toxin too? This may allow the authors to decipher the effect of lack of complete neutralization vs lack of repression.

We would like to thank this reviewer for this suggestion. We are currently in the process of a thorough structure/function analysis of the NatRT complex. This will include a detailed analysis of the flap region and its function in NatT activation and control. Given that the current manuscript is already rich in data, we strongly feel that such analyses go beyond the scope of this work.

4. The organization of Figures 4 and 4s is very confusing.

We agree with this reviewer that the arrangement of the subfigures in Fig. 4 and S4 are not linear and straightforward and may require some special attention from the reader. This is due to an attempt to graphically optimize the available space and to condense to overall figure as much as possible in light of limited publishing space.

5. Authors should better highlight the small population referred to in Figure 5b

We would like to thank this reviewer for this suggestion to make Fig. 5b more accessible to the reader. We had thought about making the small population of cells with derepressed salvage pathway genes better visible. We have refrained from adding a box to the cytogram because we have found that this mainly covers the dots and does not add to clarity. Instead, we now mention in the legend to Fig. 5b that the small subpopulation lies within the area, which is surrounded by a stippled yellow box.

“Please note the small subfraction of natTE29D mutant cells (orange dots) with increased expression of the pncB1::gfp reporter (yellow stippled box).”

Referee #2:

Santi and colleagues report an extensive structure-functional characterisation of *Pseudomonas aeruginosa* type II toxin-antitoxin pair NatR:NatT, earlier known as PA1030/PA1029 (Song et al, 2023). This is a rather understudied TA system. Song and colleagues previously used *E. coli* as a host and do a classical toxicity-neutralisation assay with wt and catalytically dead NatT. In their experiments NatT was toxic (but not very), and it was neutralised by co-expression of NatR. The weak toxicity is likely to be, as argued here by Santi and colleagues, due to rapid degradation of NatT due to poor folding.

The current study establishes that, as expected for RES toxins, NatT degrades NAD⁺. This was earlier shown for other RES-containing TAs, e.g. see (Freire et al, 2019; Skjerning et al, 2019). The authors solve X-ray structures of NatR:NatT and rationalise the gain-of-function E29D NatR mutation that results in NatT activation. This mutant was identified by the authors as associated with increased levels of antibiotic

persistence, behaving analogously to classic hipA7 allele (Korch et al, 2003). It is a solid study that opens up multiple follow-up avenues of research, with the most important being how is NatR:NatT activity induced? What is the trigger? The function of the wild-type NatR:NatT system under normal conditions is unclear, given that we do not know how the system is triggered. What does the genomic neighbourhood say? The authors mention that it is part of the core genome. Who are the neighbours?

This is a solid piece of work that uses microbiology, biochemistry and structural biology to characterise the NatR:NatT system extensively. I would recommend covering (Song et al., 2023) more extensively in the introduction - right now it is reference 30, and the toxicity-neutralisation assays from that paper are not mentioned at all. This would make the reader's life much easier. Similarly, more extensive comparison with other RES-containing TAs is warranted, e.g. (Freire et al., 2019; Skjærning et al., 2019). In general, when it comes to insights to molecular workings of RES-containing TAs, the study is adding useful new bits, but it is not ground-breaking - structures of this kind of TAs were reported before and the enzymatic activity is well-established. The interesting bit is the gain-of-function mutation of NatR that is analogous to famous hipA7 allele in terms of increased persistence (Korch et al., 2003). I would expand on that parallel - currently hipA7 is never mentioned. Overall, it is a solid piece of work but it unfortunately lacks when it comes to providing novel exciting biological insights.

We thank this reviewer for the valuable comments and for critical reading of the manuscript.

We have covered the findings by Song et al. 2023 (reference #30) early on in the introduction section of our manuscript on p.4:

Similarly, the natR-natT module of P. aeruginosa is induced under oxidative stress conditions, exposure to antibiotics and under host-like conditions²⁸⁻³⁰ and natT mutants show reduced survival during antibiotic treatment and in macrophages³⁰.

To complement this information, we have added the following sentence on p.4:

Overexpression of natT, but not expression of natR and natT together, caused a growth arrest in E. coli, indicating that NatR-NatT operates as a bona fide TA system³⁰.

We strongly disagree with the reviewer's view that our work lacks novel and exciting biological insight. The family of RES domain toxins was first described in 2019 by three different groups describing structure and (in 2 cases) biochemical activity of different members of this family (Freire et al., 2019; Skjærning et al., 2019; Piscotta et al. 2019). While Freire and co-workers demonstrated that MbcT from *M. tuberculosis* has NAD⁺ phosphorylase activity, Piscotta and colleagues showed data to suggest that ParT from *Sphingobium* sp. is an ADP ribosyl transferase. Thus, these studies failed to unambiguously answer the question of the primary activity of RES domain toxins. Also, all three publications were strongly focused on *in vitro* analyses of the respective toxins, providing only limited information about their activities *in vivo* and their specific downstream effects.

By introducing a genetically active variant of NatT, our work not only confirms that RES domain proteins are NAD⁺ phosphorylases *in vitro* and *in vivo*, but we also manage to establish a clear link between NatT-mediated NAD⁺ degradation, nutritional stress, and the formation of a small metabolically dormant subpopulation that is able to survive antibiotic treatment. Importantly, by establishing single cell readouts for NatT activity, we provide convincing evidence for the stochastic nature of the NatT toxin. The ability to monitor NatT toxin activity in individual cells is an important breakthrough in the persister and TA field as it opens up precise studies dissecting toxin activation, dormancy and persistence in greater detail. Additional key findings include the use of structural information of the inert and activated NatR-NatT complexes to provide a mechanistic frame for toxin activation and its direct coupling to transcriptional autoregulation of the *natR-natT* genes. And finally, we demonstrate that NatT activity boosts *P*.

aeruginosa virulence and provide evidence that the NatT toxin is under positive selection in patients with chronic *P. aeruginosa* infections.

Dear Urs,

Thank you for submitting a revised version of your manuscript. I sincerely apologise for the protracted assessment process due to delays in referee report submission. We have now received input from one of the original reviewers, who finds that their previous concerns have been addressed satisfactorily. Therefore, there now remain a few editorial points that need addressing before I can extend official acceptance of the manuscript:

1. Please submit up to five keywords.
2. In the Author Checklist file, please add selection in the column D.
3. Please make sure that the order of the sections in the manuscript is as follows: Title page - Abstract & Keywords - Introduction - Results - Discussion - Methods - Data Availability - Acknowledgments - Disclosure Statement & Competing Interests - References - Figure Legends - (Main Tables with legends) - Expanded View Figure Legends.
4. CRediT has replaced the traditional author contributions section because it offers a systematic, machine-readable author contributions format that allows for more effective research assessment. Please remove the Authors Contributions from the manuscript and use the free text boxes beneath each contributing author's name in our online submission system to add specific details on the author's contribution. More information is available in our guide to authors.
5. Please add a "Disclosure and competing interests statement" section, currently this information is included in the "Acknowledgments" section. (further info: <https://www.embopress.org/page/journal/14602075/authorguide#conflictsofinterest>).
6. Appendix Figure S3 currently is provided in two separate files. Since our guidelines do not allow this, please split into separate figures.
7. The Excel file with Table S1a and Table S1b sheets is a dataset that needs to be renamed to Dataset EV1; the file name, title, callouts and sheets in the Excel file need to be updated accordingly. Please remove the legend of Table S1a/S1b from the file with the other tables and include it in the Dataset Excel file, provided as a separate tab/sheet.
8. The file with supplementary tables should be part of an Appendix file - the correct nomenclature and callouts should be Appendix Table S1-S5 (or S1-S4 since the first table needs to be Dataset EV1); the Appendix file should be prefaced with a short table of contents with page numbers to show where each Appendix item is located.
9. There are currently 6 supplementary figures. We can accommodate two layers of supplementary figures. We can accommodate up to five Expanded View (EV) Figures that are collapsible/expandable online. EV Figures should be cited as 'Figure EV1, Figure EV2' etc. in the text and their respective legends should be included in the main text after the legends of regular figures. The rest of the supplementary figures should be renamed into Appendix Figure S1 etc. and added to the Appendix pdf file together with their legends.
Further information on the format is available here:
<https://www.embopress.org/page/journal/14602075/authorguide#expandedview>.
10. Please rename the movies into Movie EV1-EV2 and update the callouts accordingly. The legends should be removed from the manuscript text file and zipped with each movie file. Further information is available here:
<https://www.embopress.org/page/journal/14602075/authorguide#expandedview>
11. We require a Data Availability Section at the end of Methods, which should include resolvable links to the structural datasets. More information about the format of this section can be found here:
<https://www.embopress.org/page/journal/14602075/authorguide#dataavailability>.
12. Please update references according to The EMBO Journal style - the list should be organised alphabetically; where there are more than 10 authors on a paper, the first 10 should be listed, followed by 'et al.' Please see further information here:
<https://www.embopress.org/page/journal/14602075/authorguide#referencesformat>
13. In our standard source data check, we have noted unexplained duplicate values in the datasets for figures 1C and 2D (rows 216-217). I have attached the corresponding files with the detected duplications labelled in colour. Please check - a brief explanation would be very helpful.
14. Our data editors have flagged the following issues in figure legends that need correcting:
 - Please note that the figure 7b-c is mislabeled as figure 7c in the manuscript. This needs to be corrected.
 - Please indicate the statistical test used for data analysis in the legend of figure 2d.
 - Please note that the box plots need to be defined in terms of minima, maxima, centre, bounds of box and whiskers, and percentile in the legend of supplementary figure 4l.
 - Please provide information on the nature and number of replicates in the legends of figures 5g; 6c; 7b, supplementary figures 4c; 5g-h; 6a.
 - Please note that n=2 in figure 5e. According to our policy, calculation of SEM or other statistical information is not appropriate in case of duplicates. If possible, both data points should be shown instead.
 - Please describe the nature of replicates in the legends of figures 2a, c, e; 5e, supplementary figures 1b-c; 4a, d, g.
 - Please define the error bars in the legends of figures 5g; 6c; 7b, supplementary figures 4c; 5g-h; 6a.
15. Papers published in The EMBO Journal are accompanied online by a 'Synopsis' to enhance discoverability of the manuscript. It consists of A) a short (1-2 sentences) summary of the findings and their significance, B) 3-4 bullet points highlighting key results and C) a synopsis image that is 550x300-600 pixels large (width x height, jpeg or png format). You can either show a model or key data in the synopsis image. Please note that the image size is rather small and that text needs to be readable at the final size. Please send us this information together with the revised manuscript.

With best wishes,

Ieva

We realize that it is difficult to revise to a specific deadline. In the interest of protecting the conceptual advance provided by the work, we recommend a revision within 3 months (6th Nov 2024). Please discuss the revision progress ahead of this time with the editor if you require more time to complete the revisions.

Referee #1:

We would like to thank the authors for their detailed responses to the review comments and appreciate their efforts in addressing the points raised. Overall, the few additional experiments that are included in the revised manuscript help to bridge some of the observations that are made throughout the paper which consolidates that within the population of natT E29D, persisters, and slow growers are the same with a link with NAD salvage/levels.

We think the manuscript is ready for publication but would suggest a minor edit in the discussion section:

"Moreover, our data showed that even when NatT is unleashed, actively growing cells can neutralize its activity by replenishing NAD." The authors may want to be more explicit as they state elsewhere in the manuscript "expression and/or activity rather than by simply replenishing pools of nicotinamide dinucleotides".

Page 5 - "FLAGtagged" change to FLAG-tagged

page 10 - delta symbols are replaced with squares; this has occurred in multiple places in the manuscript.

Dear Urs,

Thank you for submitting a revised version of your manuscript. I sincerely apologise for the protracted assessment process due to delays in referee report submission. We have now received input from one of the original reviewers, who finds that their previous concerns have been addressed satisfactorily. Therefore, there now remain a few editorial points that need addressing before I can extend official acceptance of the manuscript:

1. Please submit up to five keywords.

We have added five keywords to the manuscript on page 2.

2. In the Author Checklist file, please add selection in the column D.

The column D is now completed.

3. Please make sure that the order of the sections in the manuscript is as follows: Title page - Abstract & Keywords - Introduction - Results - Discussion - Methods - Data Availability - Acknowledgments - Disclosure Statement & Competing Interests - References - Figure Legends - (Main Tables with legends) - Expanded View Figure Legends.

The sections of the manuscript were rearranged accordingly.

4. CRediT has replaced the traditional author contributions section because it offers a systematic, machine-readable author contributions format that allows for more effective research assessment. Please remove the Authors Contributions from the manuscript and use the free text boxes beneath each contributing author's name in our online submission system to add specific details on the author's contribution. More information is available in our guide to authors.

We are now using CRediT and the authors' contributions were removed from the manuscript.

5. Please add a "Disclosure and competing interests statement" section, currently this information is included in the "Acknowledgments" section. (further info:<https://www.embopress.org/page/journal/14602075/authorguide#conflictsofinterest>).

The requested section was added on page 22.

6. Appendix Figure S3 currently is provided in two separate files. Since our guidelines do not allow this, please split into separate figures.

We have transferred the second part of Figure S3 to the Appendix File (now Appendix figure S1). The first part of the figure was maintained as part of Fig. EV3.

7. The Excel file with Table S1a and Table S1b sheets is a dataset that needs to be renamed to Dataset EV1; the file name, title, callouts and sheets in the Excel file need to be updated accordingly. Please remove the legend of Table S1a/S1b from the file with the other tables and include it in the Dataset Excel file, provided as a separate tab/sheet.

Tables S1a and S1b are now called Dataset EV1. The title and callouts were updated accordingly.

8. The file with supplementary tables should be part of an Appendix file - the correct nomenclature and callouts should be Appendix Table S1-S5 (or S1-S4 since the first table needs to be Dataset EV1); the Appendix file should be prefaced with a short table of contents with page numbers to show where each Appendix item is located.

We have prepared the Appendix file accordingly.

9. There are currently 6 supplementary figures. We can accommodate two layers of supplementary figures. We can accommodate up to five Expanded View (EV) Figures that are collapsible/expandable online. EV Figures should be cited as 'Figure EV1, Figure EV2" etc. in the text and their respective legends should be included in the main text after the legends of regular figures. The rest of the supplementary figures should be renamed into Appendix Figure S1 etc. and added to the Appendix pdf file together with their legends.

Further information on the format is available

here: <https://www.embopress.org/page/journal/14602075/authorguide#expandedview>.

If possible, we would like to keep 6 Expanded view figures, each of them related to their respective main Figure (1 to 6). If not is not possible, we will transfer EV6 to the Appendix file.

10. Please rename the movies into Movie EV1-EV2 and update the callouts accordingly. The legends should be removed from the manuscript text file and zipped with each movie file. Further information is available here: <https://www.embopress.org/page/journal/14602075/authorguide#expandedview>

We have renamed the movies and updated the callouts accordingly.

11. We require a Data Availability Section at the end of Methods, which should include resolvable links to the structural datasets. More information about the format of this section can be found here: <https://www.embopress.org/page/journal/14602075/authorguide#dataavailability>.

A Data Availability section was included to the structural data.

12. Please update references according to The EMBO Journal style - the list should be organised alphabetically; where there are more than 10 authors on a paper, the first 10 should be listed, followed by 'et al.' Please see further information

here: <https://www.embopress.org/page/journal/14602075/authorguide#referencesformat>

The references have been reformatted according to the journal's style.

13. In our standard source data check, we have noted unexplained duplicate values in the datasets for figures 1C and 2D (rows 216-217). I have attached the corresponding files with the detected duplications labelled in colour. Please check - a brief explanation would be very helpful.

Figure 1C illustrates data from an antibiotic killing assay conducted under varying IPTG and cumate concentrations, which modulate NatT and NatR expression levels. Aliquots were taken before and after antibiotic treatment, and survival frequencies were calculated by dividing the post-treatment colony count by the pre-treatment count, as detailed in the Methods section. Given the experimental design, in

particular when counting dilutions with low colony numbers, identical ratios can occur across different conditions, leading to similar or duplicate values. Please note that the granularity of the figure is not affected by this, since large differences are observed in colony counts on different dilution plates.

Fig. 2D shows metabolomic data for 252 different metabolites. Indeed, two of them, dUMP and Glycineamideribotide, have the same values. This occurs because the database used for metabolite identification assigns the same mass to these two molecules when a sodium adduct is present in the Glycineamideribotide molecule. Since we cannot distinguish which of the two compounds are present, we decided to keep both of them. Importantly, these values are statistically non-significant (p -value = 2.663223, ratio NatTE29D/NatT = -0.10995) and were neither highlighted in the graph nor discussed in the manuscript, rendering them irrelevant to the study's conclusions.

14. Our data editors have flagged the following issues in figure legends that need correcting:
- Please note that the figure 7b-c is mislabeled as figure 7c in the manuscript. This needs to be corrected.

We were unable to identify mislabelings of Figure 7. In case this is still an issue, could the editor please be more specific about the location of the mislabeling in the text?

- Please indicate the statistical test used for data analysis in the legend of figure 2d.

Done

- Please note that the box plots need to be defined in terms of minima, maxima, centre, bounds of box and whiskers, and percentile in the legend of supplementary figure 4l.

Done

- Please provide information on the nature and number of replicates in the legends of figures 5g; 6c; 7b, supplementary figures 4c; 5g-h; 6a.

Done

- Please note that $n=2$ in figure 5e. According to our policy, calculation of SEM or other statistical information is not appropriate in case of duplicates. If possible, both data points should be shown instead.

Done

- Please describe the nature of replicates in the legends of figures 2a, c, e; 5e, supplementary figures 1b-c; 4a, d, g.

Done

- Please define the error bars in the legends of figures 5g; 6c; 7b, supplementary figures 4c; 5g-h; 6a.

Done

Referee #1:

We would like to thank the authors for their detailed responses to the review comments and appreciate their efforts in addressing the points raised. Overall, the few additional experiments that are included in the revised manuscript help to bridge some of the observations that are made throughout the paper which consolidates that within the population of natT E29D, persisters, and slow growers are the same with a link with NAD salvage/levels.

We thank the reviewers for the critical assessment of our manuscript that greatly improved its quality.

We think the manuscript is ready for publication but would suggest a minor edit in the discussion section:

"Moreover, our data showed that even when NatT is unleashed, actively growing cells can neutralize its activity by replenishing NAD." The authors may want to be more explicit as they state elsewhere in the manuscript "expression and/or activity rather than by simply replenishing pools of nicotinamide dinucleotides".

The first sentence refers to the activity of the salvage pathway that counteracts NatT toxicity in normal conditions. The second sentence refers to the effect of NAM addition to the medium that is capable of neutralizing NatT toxicity even in the absence of a functional salvage pathway, possible by inhibiting the activity or activation of NatT.

Page 5 - "FLAGtagged" change to FLAG-tagged

Corrected

page 10 - delta symbols are replaced with squares; this has occurred in multiple places in the manuscript.

Corrected

Dear Urs,

Thank you for addressing the final editorial issues. I am now pleased to inform you that your manuscript has been accepted for publication.

Before we forward your manuscript to our publishers, I would like to propose some minor edits in the manuscript abstract and synopsis (please see below and the attached manuscript text file). I have also written a short blurb that will accompany the title of your manuscript in our online system. Please let me know if any corrections or adjustments are needed.

Blurb:

The toxin NatT promotes drug tolerance in *Pseudomonas aeruginosa* under nutrient-limited conditions and is activated in clinical isolates.

Synopsis:

The toxin-antitoxin system NatR-NatT promotes persister formation in *Pseudomonas aeruginosa*. This study shows that NatT acts by depleting the essential co-factors NAD and NADP and promotes drug tolerance in this human pathogen.

- The RES domain of NatT degrades NAD⁺ into nicotinamide and ADP-ribose-1'-phosphate.
- Actively growing bacteria can counteract NAD⁺ depletion by inducing the NAD⁺ salvage pathway.
- Under nutrient-limited conditions, bacteria are unable to compensate for NatT-mediated NAD⁺ depletion, resulting in dormancy and persistence.
- NatT toxin activation is inhibited by the NAD⁺ precursor nicotinamide.

If you have any questions, please do not hesitate to contact the Editorial Office. Thank you for this contribution to The EMBO Journal and congratulations on a great study!

Best wishes,

Ieva
